# BO4Mob: Bayesian Optimization Benchmarks for High-Dimensional Urban Mobility Problem

**Seunghee Ryu**[1,2*]   **Donghoon Kwon**[1,2*]   **Seongjin Choi**[1†]
**Aryan Deshwal**[1]   **Seungmo Kang**[2]   **Carolina Osorio**[3]
[1]University of Minnesota   [2]Korea University   [3]HEC Montréal
{sryu, dkwon, chois, adeshwal}@umn.edu
s_kang@korea.ac.kr
carolina.osorio@hec.ca

## Abstract

We introduce **BO4Mob**, a new benchmark framework for high-dimensional Bayesian Optimization (BO), driven by the challenge of origin-destination (OD) travel demand estimation in large urban road networks. Estimating OD travel demand from limited traffic sensor data is a difficult inverse optimization problem, particularly in real-world, large-scale transportation networks. This problem involves optimizing over high-dimensional continuous spaces where each objective evaluation is computationally expensive, stochastic, and non-differentiable. BO4Mob comprises five scenarios based on real-world San Jose, CA road networks, with input dimensions scaling up to 10,100. These scenarios utilize high-resolution, open-source traffic simulations that incorporate realistic nonlinear and stochastic dynamics. We demonstrate the benchmark's utility by evaluating five optimization methods: three state-of-the-art BO algorithms and two non-BO baselines. This benchmark is designed to support both the development of scalable optimization algorithms and their application for the design of data-driven urban mobility models, including high-resolution digital twins of metropolitan road networks. Code and documentation are available at `https://github.com/UMN-Choi-Lab/BO4Mob`.

## 1   Introduction

### 1.1   Motivation

Cities worldwide are increasingly developing digital twins of their urban mobility systems. This is due to the increasing complexity of the systems: numerous stakeholders (e.g., travelers, public and private sector mobility service operators, governmental agencies), numerous interacting mobility services (e.g., on-demand ride-sharing, public transportation services), and the shift towards dynamic (e.g., real-time, time-dependent) service operations (e.g., surge pricing for on-demand services, dynamic congestion pricing, real-time speed limits, traffic-responsive traffic signal strategies).

Urban mobility digital twins rely on a high-resolution description of traffic dynamics provided by what are known as **traffic simulators**. These simulators describe the behavior of both demand and supply in detail. On the demand side, individual vehicles have their own technology (e.g., electric, autonomous, connected), and individual travelers have their own behavior (e.g., aggressive drivers, business travelers with a high value of time, different willingness to shift to new travel modes). On the supply side, the operations of the city's infrastructure (e.g., traffic signal plans, congestion pricing policies, fleet of public transportation vehicles) and of the available mobility services (e.g.,

---

*Work done while at University of Minnesota. Equal contribution
†Corresponding Author

39th Conference on Neural Information Processing Systems (NeurIPS 2025) Track on Datasets and Benchmarks.

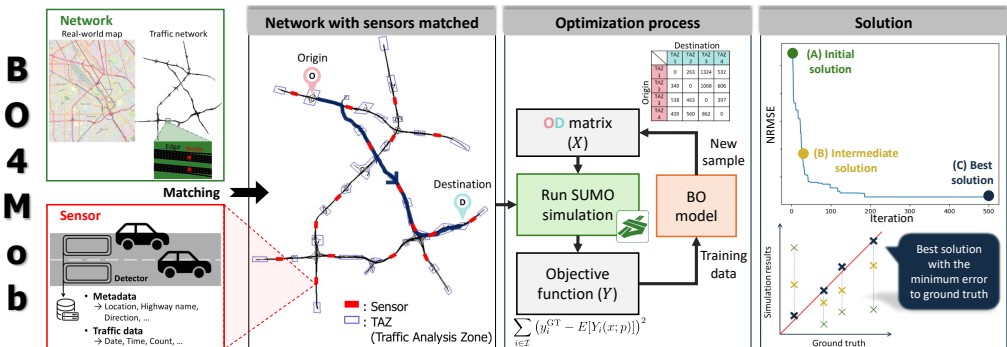

Figure 1: Illustration of the overall workflow for OD estimation using BO, where sensor data and traffic simulation are used to evaluate the quality of candidate OD demands and guide the optimization process.

taxi offerings, bike-sharing, on-demand ride-hailing) are also modeled in detail. The underlying demand models that govern the choices of travelers such as mode choice, departure time choice, route choice, and lane choice, are most often based on well-established probabilistic models (e.g., random utility models). Hence, the realization of the trip of an individual involves sampling from a number of probability distributions. The resulting simulation of the urban mobility system then involves simulating the trips of a large (e.g., tens or hundreds of thousands) population of vehicles and travelers. **This combination makes the simulator inherently stochastic, and computationally expensive.** For example, simulating a large network with tens to hundreds of thousands of agents can take up to 10 hours. Moreover, the simulator's mapping of inputs (e.g., travel demand) to outputs (e.g., traffic and congestion statistics) is non-differentiable.

As a result, optimization tasks involving these simulators naturally fall under the category of **high-dimensional black-box optimization with a computationally expensive oracle.** Evaluating a candidate solution (e.g., demand profile or policy intervention) requires a compute-costly simulation run. Moreover, transportation budget and operational constraints can lead to intricate feasible regions defined by nonlinear inequality constraints, or even simulation-based constraints. Hence, there is potential for the research communities of sample-efficient optimization, and more specifically Bayesian optimization (BO), to contribute to advancing the science and the practice of urban mobility planning and operations. Especially, recent advances in high-dimensional BO have introduced techniques that are well-suited for problems like origin-destination (OD) travel demand estimation in large-scale urban road networks, where the objective is to estimate travel demand (OD matrix) between origin and destination pairs that best matches the observed traffic counts on road links throughout the network. **Despite its potential, there is no established benchmark connecting BO methods to realistic urban mobility problems.** This lack of standardization hinders both methodological progress and practical adoption.

## 1.2 Contributions

This paper discusses the, arguably, most important and difficult open optimization problem in the development of digital twins for urban mobility systems. Specifically, we present a new reproducible real-world benchmark framework for high-dimensional OD travel demand estimation, also referred to as *OD estimation* or *OD calibration*, using BO as illustrated in Figure 1. We refer to this benchmark as **BO4Mob**. Our main contributions are as follows.

- The proposed BO4Mob bridges black-box optimization and transportation engineering by introducing a standardized benchmark that brings BO research closer to impactful real-world mobility problems and inspires further development within the BO community.

- The benchmark includes five road network instances (see Figure 2 and Table 1) that are the building blocks of any freeway network. These instances are constructed with increasing levels of complexity, enabling systematic evaluation across different scales, which is a particularly time-intensive and uncommon effort in this field.

- We implement realistic traffic scenarios using the open-source SUMO simulator [Lopez et al., 2018], which models nonlinear and stochastic dynamics, and incorporate real-world sensor placements and observation periods to reflect actual urban sensing conditions.

- All datasets, simulation setups, and evaluation tools are released as open-source to promote reproducibility and extensibility. This level of integration remains uncommon in transportation research despite relying on public data sources.

## 2 Related work

### 2.1 Bayesian optimization benchmarks

Existing high-dimensional BO methods rely on a relatively narrow set of benchmark families, limiting the breadth of empirical insights that can be obtained. One such category of benchmarks comprises analytic global optimization test functions. Canonical examples in this category include analytic functions like Ackley, Rastrigin, Styliblinksi-Tang, together with the broader Black-Box Optimization Benchmarking (BBOB) suite [Binois and Wycoff, 2022, Finck et al., 2010]. Despite having the nice property of being inexpensive to evaluate, these functions are unrealistic proxies of real-world optimization problems. Their standard form contains no noise, constraints, or complex inter-dependencies inherent in real-world problems. A second widely adopted evaluation setting in BO papers revolves around hyperparameter optimization of machine learning models [Eggensperger et al., 2021]. However, these benchmarks are usually restricted to fewer than 10 dimensions and focus primarily on mixed (discrete-continuous) and conditional search space structures. Consequently, their utility for stress-testing scalability in the high-dimensional continuous space regime is limited. LassoBench [Šehić et al., 2022] is a recent benchmark for hyperparameter optimization in weighted lasso regression. While this benchmark contains problems over high-dimensional inputs, it has been shown in recent work that only a small subset of the input variables significantly influence the objective value [Papenmeier et al., 2025]. Recently, poli [González-Duque et al., 2024] introduced a set of high-dimensional benchmarks for protein and small-molecule optimization tasks, but they focus only on discrete sequences.

### 2.2 High-dimensional Bayesian optimization

BO becomes particularly challenging in high-dimensional settings, which typically involve more than 20 variables, due to the curse of dimensionality [Frazier, 2018, Nayebi et al., 2019]. Although recent work has scaled BO to problems with hundreds or even up to 20,000 dimensions [Zhang et al., 2019], surrogate models like Gaussian processes often lose predictive accuracy and become harder to optimize as dimensionality increases, which reduces sample efficiency.

To mitigate these challenges, various methods modify the optimization procedure or impose structural assumptions such as sparsity, decomposability, or locality. Notable approaches include additive models [Kandasamy et al., 2015, Rolland et al., 2018], subspace embeddings [Wang et al., 2016, Letham et al., 2020, Papenmeier et al., 2022], sparse-input priors [Eriksson and Jankowiak, 2021], trust region methods [Eriksson et al., 2019], latent-variable models [Oh et al., 2018], and neural surrogates [Springenberg et al., 2016]. The effectiveness of these methods depends on how well their assumptions reflect the structure of the problem. However, recent studies suggest such assumptions may not be necessary. With proper prior scaling, robust initialization, and simple local search strategies, standard BO can remain competitive in thousands of dimensions [Hvarfner et al., 2024, Xu et al., 2025a, Papenmeier et al., 2025].

### 2.3 Urban mobility problems & OD estimation

Urban mobility systems involve many interacting agents, services, and infrastructure, making them complex and dynamic. In response, digital twins have emerged as a promising framework for real-time monitoring, prediction, and decision-making by maintaining a continuous feedback loop between the physical network and its virtual representation [Jones et al., 2020]. OD estimation is a key component of these systems, providing demand inputs for traffic simulation [Kušić et al., 2023]. Recent studies have explored deep learning approaches [Min et al., 2024] and hybrid simulation models that combine microscopic and macroscopic layers [Xu et al., 2025b].

Because OD demand cannot be directly observed from sparse and noisy traffic data, the task becomes a high-dimensional, under-determined inverse optimization problem. To improve sample efficiency, recent methods incorporate structural or analytical priors. For instance, Osorio [2019] propose a metamodel-based approach that embeds analytical models to reduce simulation cost. Other approaches leverage macroscopic relationships, such as the macroscopic fundamental diagram (MFD), or integrate data sources like probe vehicle and transit data [Dantsuji et al., 2022].

# 3 Bayesian optimization benchmark for high-dimensional urban mobility problem

## 3.1 Preliminaries - Bayesian Optimization

BO is a sample-efficient global optimization strategy for solving black-box functions that are expensive to evaluate. Formally, given an unknown objective function $f : \mathcal{X} \to \mathbb{R}$ defined over a compact domain $\mathcal{X} \subset \mathbb{R}^D$, the goal is to find

$$\mathbf{x}^\star = \arg \min_{\mathbf{x} \in \mathcal{X}} f(\mathbf{x}), \tag{1}$$

under the constraint that $f$ can only be queried point-wise and is expensive to evaluate (e.g., through simulation or experiment), and that gradients are unavailable.

BO constructs a probabilistic surrogate model, typically a Gaussian Process (GP), to model the distribution over functions:

$$f(\mathbf{x}) \sim \mathcal{GP}(\mu(\mathbf{x}), k(\mathbf{x}, \mathbf{x}')), \tag{2}$$

where $\mu(\cdot)$ is the mean function and $k(\cdot, \cdot)$ is the kernel function capturing the correlation between function values.

An acquisition function $\alpha : \mathcal{X} \to \mathbb{R}$ is then used to guide the selection of the next query point by balancing exploration and exploitation:

$$\mathbf{x}_{t+1} = \arg \max_{\mathbf{x} \in \mathcal{X}} \alpha(\mathbf{x}; \mathcal{D}_t), \tag{3}$$

where $\mathcal{D}_t = \{(\mathbf{x}_i, f(\mathbf{x}_i))\}_{i=1}^t$ is the set of observed data. Common acquisition functions include Expected Improvement (EI), Upper Confidence Bound (UCB), and Probability of Improvement (PI).

This iterative process continues until a budget constraint (e.g., number of queries or computational time) is reached.

## 3.2 Problem formulation

In OD estimation problems, a metropolitan area is typically divided into traffic analysis zones (TAZes), and the goal is to infer a high-dimensional vector of travel demands between selected OD pairs so that simulated traffic statistics align closely with observed field data. The number of OD pairs determines the dimensionality of the problem.

The problem can be formulated as follows,

$$\min_{\mathbf{x} \in \Omega} \sum_{i \in \mathcal{I}} \left( y_i^{\mathrm{GT}} - \mathbb{E}[Y_i(\mathbf{x}; \mathbf{p})] \right)^2, \tag{4}$$

where $y_i^{\mathrm{GT}}$ denotes a statistic obtained from ground-truth (GT) (i.e., field) traffic data, such as average traffic count, speed, travel time, on a given link $i$, and $\mathbb{E}[Y_i(\mathbf{x}; \mathbf{p})]$ denotes the simulated counterpart derived from the traffic simulator. The latter depends on the vector of travel demands $\mathbf{x}$, a vector of additional parameters $\mathbf{p}$ (e.g., road attributes such as number of lanes, speed limits, etc.), and the set of link indices in the network where traffic data or measurements are available, denoted $\mathcal{I}$. In its simplest form, the feasible region $\Omega$ consists of simple upper and lower bound constraints.

This formulation aims to minimize the squared distance between the historical traffic statistics and the corresponding simulated counterparts. The inverse optimization community will recognize Problem (4) as an inverse optimization problem. The challenges of tackling this problem are: (i) $\mathbf{x}$ is high-dimensional (upto tens or hundreds of thousands), (ii) $\mathbb{E}[Y_i(\mathbf{x}; \mathbf{p})]$ is an unknown, costly to compute, function that can only be estimated via stochastic simulation.

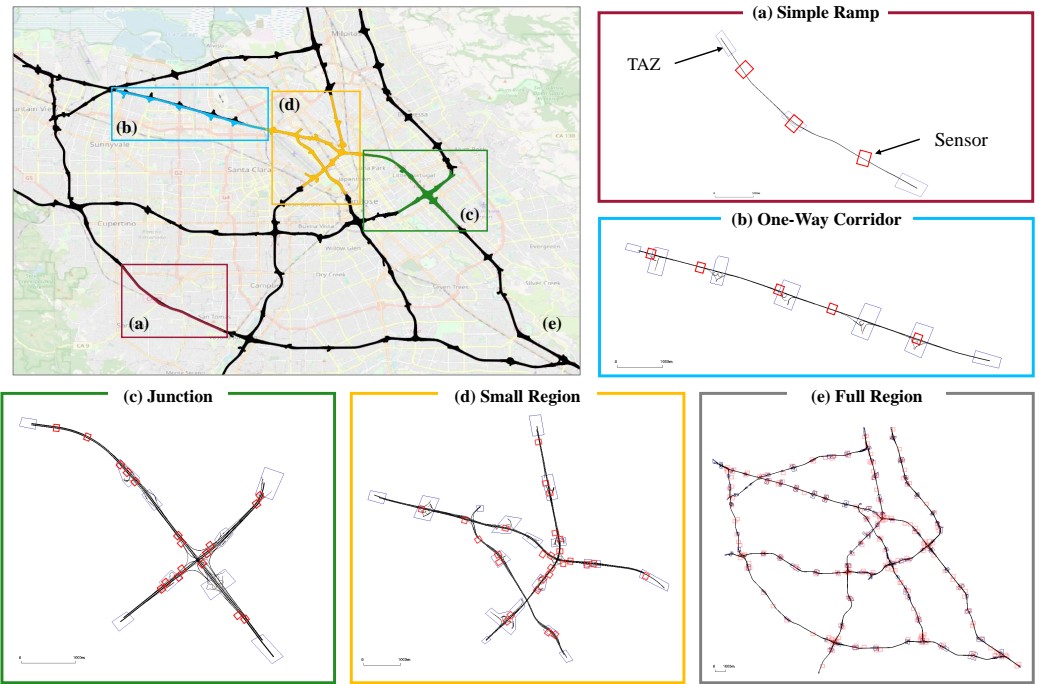

Figure 2: Freeways extracted from the San Francisco Bay Area network, including a total of eight freeways. To evaluate the performance of BO algorithms, five subnetworks are utilized: (a) Simple Ramp, (b) One-Way Corridor, (c) Junction, (d) Small Region, and (e) Full Region.

## 3.3    Benchmark networks

To systematically evaluate algorithmic performance at varying scales of complexity, we extract five subnetworks from the San Francisco Bay Area network. Figure 2 provides a visual overview of the subnetworks, and Table 1 lists their key characteristics (e.g., number of nodes, links, TAZes, OD pairs ($\mathbf{x}$), sensors). Below, we summarize the design rationale and main features of each subnetwork.

- **Simple Ramp:** This network is the smallest and simplest subnetwork that consists of a small linear topology network and single pair of ramps (one on-ramp and one off-ramp). This problem is the only deterministic case. This serves as a test environment to validate modeling assumptions, parameter settings, and/or initial algorithmic prototypes before scaling up.

- **One-Way Corridor:** This subnetwork offers an unidirectional freeway corridor with multiple links and ramps. Although still moderate in complexity, it introduces overlapping OD pairs and ramp interactions. This network tests the robustness of calibration or optimization algorithms in scenarios where congestion may propagate over several links.

- **Junction:** This network represents a freeway junction where two major freeways intersect, connected via both on-/off-ramps and direct freeway-to-freeway connectors. The merging and diverging flows create complex local interactions, making it a suitable scenario to test whether calibration algorithms can handle localized congestion, sharp flow transitions, and flow redistribution across multiple paths.

- **Small Region:** This network spans a broader area than the previous subnetworks and includes multiple corridors and junctions. Compared to more localized settings, it features more distributed OD demand and increased interaction across different parts of the network. This setting is well-suited for evaluating the scalability of calibration algorithms under higher spatial complexity.

- **Full Region:** This network encompasses the entire freeway system of the region, covering all major corridors and junctions. It involves high-dimensional OD demand and complex spatial interactions across a large urban area. This setting presents a realistic and challenging scenario for evaluating the generalization and efficiency of calibration algorithms at full scale.

Table 1: Network size and simulation run time (average and standard deviation) computed over 20 simulation runs.

| Type | Size of network | | | | | Simulation run time (sec) | |
|---|---|---|---|---|---|---|---|
| | Nodes | Links | TAZes | OD pairs (x) | Sensors (y) | Avg | SD |
| Simple Ramp | 10 | 10 | 3 | 3 | 3 | 0.80 | 0.29 |
| One-Way Corridor | 66 | 68 | 7 | 21 | 5 | 5.94 | 1.34 |
| Junction | 137 | 152 | 9 | 44 | 18 | 11.88 | 2.70 |
| Small Region | 251 | 270 | 16 | 151 | 27 | 82.72 | 7.53 |
| Full Region | 1,977 | 2,173 | 101 | 10,100 | 219 | 40,099.05 | 1,555.37 |

## 3.4 Data

As shown in Table 1, our benchmark includes five network configurations of varying scales and complexity: Simple Ramp, One-Way Corridor, Junction, Small Region, and Full Region. The simulation network is based on the preprocessed SUMO traffic simulation model provided in the Supplementary Note 2 of Ambühl et al. [2023].[3] For this benchmark, we extract the freeway network centered near San Jose, CA. Each network differs in the number of nodes, links, TAZes, OD pairs, and sensors.

To provide realistic and reproducible traffic count observations, we incorporate **real-world traffic detector data** from the Caltrans Performance Measurement System (PeMS)[California Department of Transportation]. This data is often called PEMS-BAY data and is widely used as short-term traffic forecasting benchmark data [Wu et al., 2019, Choi et al., 2025]. PeMS collects extensive traffic state data across the state of California using double loop detectors, including traffic counts (or traffic count; the number of vehicles passing a given location during a fixed time interval) and traffic speeds averaged over 5-minute intervals. While our main analysis focuses on a representative day, we also provide traffic count and average speed data from various dates and other hourly periods to support extensibility of the analysis.[4]

These traffic count measures are used as the GT values ($y_i^{GT}$) in Equation (4). We focus exclusively on "Main Line" (ML) detectors that are placed along freeway links rather than on ramp links.

To align the sensor data with the simulation network, we perform a sensor-to-link matching process. Each sensor is mapped to the closest freeway link in the network using metadata including its location and freeway travel direction. This ensures that observed traffic counts are accurately assigned to corresponding network links with consistent spatial and directional alignment. In Table 1, the *Sensors* column lists the number of sensors retained after applying data quality and spatial resolution filtering. The detailed filtering procedures are described in Appendix B.

## 3.5 Baseline methods

To assess the relevance and difficulty of the proposed benchmark, we evaluate a set of optimization methods that represent a variety of approaches to high-dimensional black-box optimization. The selection includes both a classical baseline and recent methods that incorporate structural assumptions, as well as a simple random search baseline for reference.

- **Simultaneous Perturbation Stochastic Approximation (SPSA) [Spall, 1992, 2005]:** A standard non-BO baseline widely used in OD calibration tasks [Osorio, 2019, Dantsuji et al., 2022]. SPSA approximates gradients via simultaneous perturbations across all input dimensions, requiring only two function evaluations per iteration. This makes it well-suited for simulation-based optimization under limited evaluation budgets.

- **Vanilla Bayesian Optimization (Vanilla BO) [Jones et al., 1998]:** A standard BO baseline without explicit structural assumptions. It provides a reference point for evaluating the impact of structure-aware models in high-dimensional spaces.

---

[3]https://www.research-collection.ethz.ch/handle/20.500.11850/584669
[4]preprocessing code available at https://github.com/UMN-Choi-Lab/PeMS-BAY-2022

- **Sparse Axis-Aligned Subspace Bayesian Optimization (SAASBO) [Eriksson and Jankowiak, 2021]:** Incorporates sparsity-inducing priors in the surrogate model to identify and focus on the most relevant input dimensions, improving efficiency in high-dimensional settings.
- **Trust Region Bayesian Optimization (TuRBO) [Eriksson et al., 2019]:** Performs BO within adaptive trust regions, enabling scalable optimization in high dimensions by focusing on local modeling.

All BO baseline methods were implemented using the BoTorch framework [Balandat et al., 2020], following or slightly adapting available tutorials [BoTorch]. Detailed implementation settings are provided in Appendix C.

## 4 Experiments

### 4.1 Evaluation metric

To assess the accuracy of the estimated OD $\mathbf{x}$, we employ the normalized root mean squared error (NRMSE) between simulated and GT traffic counts across all links with sensors $\mathcal{I}$. NRMSE is defined as:

$$\text{NRMSE}(\mathbf{x}) = \frac{\sqrt{\frac{1}{n} \sum_{i \in \mathcal{I}} \left( y_i^{\text{GT}} - y_i^{\text{sim}}(\mathbf{x}) \right)^2}}{\frac{1}{n} \sum_{i \in \mathcal{I}} y_i^{\text{GT}}}, \tag{5}$$

where $y_i^{\text{sim}}(\mathbf{x})$ represents the simulated traffic counts based on the OD estimate $\mathbf{x}$, it is as an estimate of $\mathbb{E}[Y_i(\mathbf{x}; \mathbf{p})]$ in Problem (4), while $n = |\mathcal{I}|$ denotes the number of links with sensors. This formulation allows for consistent comparison of estimation quality across networks of varying scales.

To highlight the performance gain achieved by each optimization method, we compute the percentage improvement:

$$\text{Improvement}(\mathbf{x}^{\text{best}}) = \frac{\text{NRMSE}(\mathbf{x}_{\min}^{\text{init}}) - \text{NRMSE}(\mathbf{x}^{\text{best}})}{\text{NRMSE}(\mathbf{x}_{\min}^{\text{init}})} \times 100\%, \tag{6}$$

where $\mathbf{x}^{\text{best}} = \arg\min_{\mathbf{x} \in \mathcal{X}_{\text{eval}}} \text{NRMSE}(\mathbf{x})$ denotes the best-performing solution within a single optimization run, selected among all candidates evaluated throughout the process. Here, $\mathcal{X}_{\text{eval}} = \mathcal{X}_{\text{init}} \cup \bigcup_{t=1}^{T} \mathcal{X}_t$, where $\mathcal{X}_{\text{init}}$ is the set of initial candidate solutions, $\mathcal{X}_t$ is the set of solutions evaluated at epoch $t$, and $T$ is the total number of epochs. The initialization baseline $\mathbf{x}_{\min}^{\text{init}} = \arg\min_{\mathbf{x} \in \mathcal{X}_{\text{init}}} \text{NRMSE}(\mathbf{x})$ is shared across all methods. This metric quantifies the relative reduction in error achieved by each optimization model from a common initialization.

### 4.2 Simulation setup

Traffic simulations were conducted using the SUMO traffic simulator. For each candidate OD, we generated trip files based on TAZes, applied predefined routing adjustments to expedite simulation runs, and conducted full network simulations to produce simulated traffic counts on links with sensors. Simulations were executed in mesoscopic simulation mode to accelerate computation while preserving traffic count fidelity at the network level. To ensure reproducibility, each simulation run used a fixed random seed.

Table 1 reports the corresponding simulation run times for each network, highlighting the increasing computational cost as network size grows. Simulation time scales sharply with network size, ranging from under one second for the Simple Ramp network to over 40,000 seconds for the Full Region network. All simulation run times were measured on a high-performance server equipped with dual AMD EPYC Milan 7643 CPUs (96 cores total), and 1TB DDR4 memory. SUMO simulations were executed without GPU acceleration, utilizing up to six parallel processes via Python's multiprocessing module. More detailed simulation configurations, including simulation time windows and sensor observation periods, are summarized in Appendix D.

### 4.3 Analysis results

**Enhancing OD estimation via Bayesian optimization**  Table 2 reports average $\text{NRMSE}(\mathbf{x}^{\text{best}})$ and $\text{Improvement}(\mathbf{x}^{\text{best}})$ across multiple independent runs per network. The average NRMSE of the

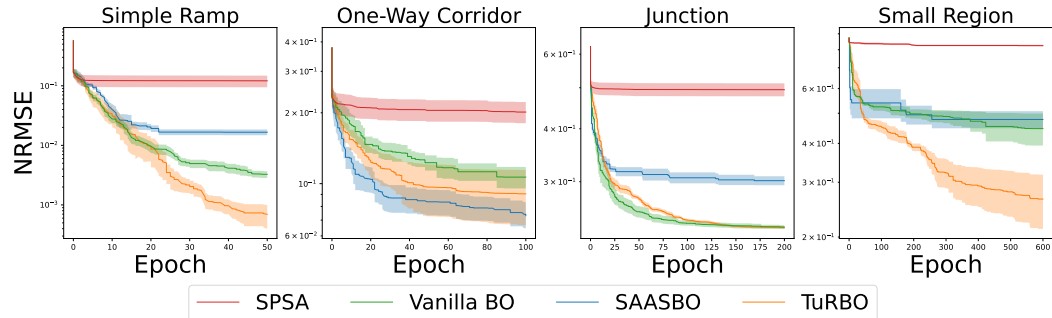

Figure 3: Convergence behavior across optimization methods. NRMSE is plotted on a logarithmic scale. Solid lines represent the mean, with error bars denoting one standard deviation over multiple runs.

initial solution is also reported for reference. Improvements are computed per run as the relative reduction from the best initial candidate to the best final solution. Run counts and optimization budgets per network are listed in Table 3. Initial candidate pools were independently sampled per run and shared across methods to ensure fairness. BO methods consistently outperform random search and SPSA, confirming the benefit of surrogate modeling in noisy, high-dimensional settings. These results underscore BO's potential as a sample-efficient framework for OD estimation across diverse networks.

**Impact of network complexity** As network size increases, OD calibration problem tends to become more difficult. The minimum NRMSE among initial candidates rises from 0.167 (Simple Ramp) to 0.847 (Small Region). After optimization, NRMSE tends to remain higher and improvements smaller on complex networks, suggesting reduced optimization leverage. For the Full Region, only a few runs per method were conducted with reduced optimization settings due to high computational cost (over 40,000 seconds per simulation; see Table 1), but all four models did not yield any performance improvement. These results highlight that OD calibration remains increasingly challenging in high-dimensional settings, emphasizing the need for scalable and sample-efficient optimization approaches.

**Comparative method analysis** Figure 3 summarizes the convergence behaviors of each method across networks, with the x-axis denoting epochs and the y-axis showing NRMSE on a logarithmic scale. The figure aggregates multiple runs per method, where the solid line shows the mean and shaded areas indicate variability across seeds. The corresponding final NRMSE values are reported in Table 2. Applying existing optimization methods to the OD calibration benchmarks, we observe that TuRBO generally delivers the strongest performance. It achieves the best results on the Simple Ramp and Small Region networks, and performs comparably to the top method on the other networks, with only minor differences in best NRMSE and convergence profiles. Vanilla BO performs competitively, especially on mid- to large-scale networks such as Junction and Small Region, where it outperforms SAASBO. SAASBO shows its strongest performance on the One-Way Corridor network, but tends to

Table 2: Average NRMSE of the initial solution is reported for reference, followed by the average best NRMSE values across five network types and five optimization models. Values in parentheses indicate the average percentage improvement. Both averaged across independent runs.

| Network* | Initial solution | | Model | | | | |
|---|---|---|---|---|---|---|---|
| | Avg | Min | Random search | SPSA | Vanilla BO | SAASBO | TuRBO |
| Simple Ramp | 0.396 | 0.167 | 0.071 (57.43%) | 0.121 (27.10%) | 0.003 (97.81%) | 0.017 (88.87%) | **0.001 (99.47%)** |
| One-Way Corridor | 0.524 | 0.316 | 0.145 (37.83%) | 0.196 (15.61%) | 0.105 (53.12%) | **0.07 (68.56%)** | 0.09 (61.25%) |
| Junction | 0.636 | 0.509 | 0.436 (14.29%) | 0.495 (2.83%) | **0.233 (54.15%)** | 0.302 (40.54%) | 0.234 (53.96%) |
| Small Region | 0.882 | 0.847 | 0.598 (29.43%) | 0.825 (2.59%) | 0.437 (48.50%) | 0.477 (43.60%) | **0.258 (69.54%)** |

*Full Region was evaluated under a reduced optimization setting (see Table 3), but all four models did not yield any performance improvement.

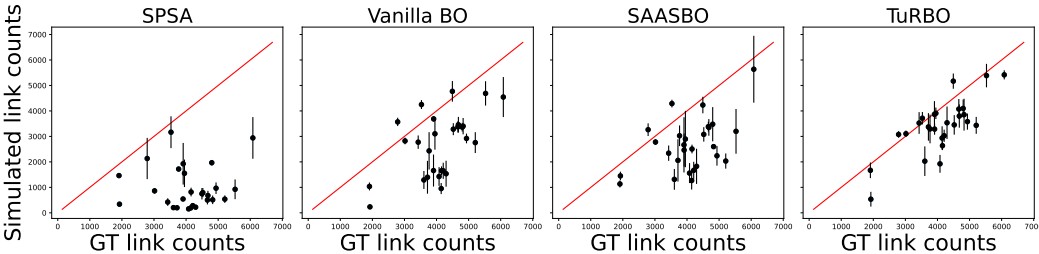

Figure 4: Fit to GT link traffic counts for the Small Region network. Each black dot represents the mean of GT link traffic counts corresponding to the OD that yielded the minimum loss for a given random seed. Error bars denote one standard deviation across seeds.

underperform on more complex scenarios. In contrast, random search and SPSA consistently show limited convergence across networks, regardless of size.

**Alignment between simulated and ground-truth traffic counts**    To assess the quality of OD calibration, we compare the simulated traffic counts generated from the optimized OD $\mathbf{x}^{\text{best}}$ with the observed GT traffic counts. Figure 4 shows the alignment between simulated and GT traffic counts on the Small Region network, where the x-axis represents GT link traffic counts and the y-axis represents simulated link traffic counts. The diagonal line indicates perfect agreement. TuRBO achieves the closest alignment to GT traffic counts, producing near-linear correspondence across a wide range of traffic counts. Vanilla BO and SAASBO show similar alignment patterns, both exhibiting underestimation across most sensors. SPSA exhibits more pronounced underestimation and larger deviations overall, reflecting its limited optimization effectiveness. Additional fit plots for other networks are provided in Appendix E.

## 5   Discussion

In the transportation literature, it is well established that there is a need to develop sample-efficient methods to tackle the high-dimensional real-world OD estimation instances. The communities of black-box optimization, and particularly that of BO, are well-equipped to contribute to this challenge. Our benchmark, BO4Mob, is designed to support future research by providing a realistic testbed for advancing BO techniques inspired by challenges in urban transportation systems. We describe multiple future research directions that can leverage BO4Mob below.

Impactful research directions include the formulation and use of uncertainty quantification methods for OD estimation. Since OD estimation is an under-determined problem (i.e., there is a continuum of ODs that fit equally well the field traffic data), sensor noise and missing measurements further distort the objective landscape, making it difficult for BO to recover the true OD matrix. It is therefore important to account for this input uncertainty when using the simulation models for downstream tasks (e.g., to evaluate the impact of the deployment of a future transportation policy, such as traffic management or congestion pricing). This would enable transportation practitioners to use the simulators to perform robust counterfactual analysis. Similarly, uncertainty quantification methods allow us to quantify the level of under-determination of the instance. This can also be used to identify optimal sensor placements, guiding transportation practitioners on the type and location of new sensors such as to reduce the level of under-determination, and thereby increase counterfactual robustness.

A major challenge of standard BO methods is their lack of scalability: they do not perform well for high-dimensional instances. There is a wealth of suitable (e.g., inexpensive to compute, differentiable) surrogate models that stem from the transportation literature that can be used to embed traffic physics and capture the strong spatial dependencies inherent to transportation networks, thereby providing structural information for a black-box solver. These surrogates can be used to enhance the scalability of standard BO techniques by designing physics-informed kernels for BO [Tay and Osorio, 2022] or to physics-informed prior function distributions. The performance of standard BO solvers is highly sensitive to the initial sample. The use of traffic-physics to define tailored sampling mechanisms, such as in Tay and Osorio [2024], is a promising research direction that becomes particularly important in

high-dimensional instances. Such advancements in sample efficiency can now be rigorously tested with BO4Mob.

Additionally, the loss function of Equation (4) maps to a scalar. In order to exploit more detailed information from the expensive-to-run simulations, one can consider multi-output formulations where each term in the sum of squares is an output. In this case, multi-output GPs can be an interesting modeling choice, and one can resort to existing BO solvers that can tackle multi-dimensional outputs. This multi-output modeling becomes particularly relevant for time-dependent OD estimation problems (known as dynamic OD estimation problems), where the goal is to estimate a time-ordered sequence of ODs, rather than a single OD. This allows for the description of the temporal variations in travel demand.

Beyond methodological enhancements, the benchmark itself is designed to support easy extensions. Multi-objective formulations can incorporate additional metrics such as average link travel speed or travel time alongside link traffic counts. Demand can also be stratified by transportation mode (e.g., separate OD demand for trucks, buses, and passenger cars). The framework generalizes to a wide range of network types and sizes, supporting its applicability to diverse urban contexts. These directions open up opportunities for both more realistic modeling and broader benchmarking.

## 6    Conclusion

We present a benchmark suite for high-dimensional OD estimation based on realistic traffic simulation models. OD estimation is a core yet challenging component of urban mobility modeling, particularly in large-scale networks where the dimensionality of the problem makes direct optimization intractable. OD estimation in such systems is high-dimensional (up to thousands of variables), non-differentiable due to the stochastic nature of simulators, black-box and computationally expensive (e.g., a Full Region simulation takes over 11 hours), and under-determined since multiple ODs may explain the same sensor data. Such characteristics closely match the challenges that BO is designed to address, as it builds sample-efficient surrogate models, selects candidates through acquisition functions, and effectively handles noisy and black-box objectives. However, evaluating BO under these realistic and computationally demanding conditions has been difficult due to the lack of standardized benchmarks that connect optimization methods with real-world transportation models. Our benchmark fills this gap by integrating real-world sensor data, scalable network instances, and BO methods to evaluate sample efficiency in complex urban mobility settings. Empirical results show that BO methods outperform conventional baselines, though performance degrades as dimensionality increases. The Full Region case in particular highlights the computational and methodological challenges of scaling BO to urban-scale problems.

## Acknowledgements

Seunghee Ryu and Donghoon Kwon are affiliated with the Department of Civil, Environmental and Architectural Engineering, Korea University, South Korea, and are supported by the Basic Science Research Program through the National Research Foundation of Korea (NRF), funded by the Ministry of Education, South Korea (RS-2020-NR049594), and by the BK21 FOUR (Brain Korea 21 Four) Project; Support Program for Outstanding Graduate Students' International Joint Training. Seongjin Choi is supported by the Department of Civil, Environmental and Geo-Engineering and Center for Transportation Studies at the University of Minnesota. Seungmo Kang is supported by the Basic Science Research Program through the NRF, funded by the Ministry of Education, South Korea (RS-2020-NR049594).

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

# A  OD pair generation

For each benchmark network, OD pairs are generated based on a partitioning of the network into TAZes. Each TAZ may contain source nodes, sink nodes, or both (see Figure 5). A TAZ with at least one source node can serve as an origin, and one with at least one sink node can serve as a destination. To ensure full OD connectivity, we construct TAZes such that each zone contains at least one source node and at least one sink node, except in one-way networks (e.g., Simple Ramp, One-Way Corridor) where directional flow constraints may lead some TAZes to contain only sources or only sinks. This guarantees that every TAZ can act as both an origin and a destination. The granularity of the TAZ configuration can be adjusted: finer partitions allow for detailed spatial demand analysis, while coarser ones support higher-level planning.

Let $\tau$ denote the number of TAZes in a given network. Once TAZes are defined, OD pairs are constructed by treating each TAZ as a potential origin and destination. Since our benchmark adopts relatively fine-grained TAZ definitions, we assume that intra-TAZ trips (i.e., trips with the same origin and destination zone) are excluded. Under this assumption, the theoretical maximum number of OD pairs is $\tau(\tau - 1)$.

In practice, not all OD pairs are feasible due to network topology constraints. In some subnetworks, there may exist no valid path between certain origin and destination TAZes under the given routing assumptions. Figure 6 illustrates such a case: `taz_85` includes two source nodes and one sink node, but due to road directionality and the limited network extent, trips from `taz_85` can only reach `taz_1`, and trips to `taz_85` can only originate from `taz_1`. As a result, OD pairs involving other zones such as `taz_31` are infeasible. To systematically identify and exclude such pairs, we use the *od2trips* tool from SUMO, which attempts to generate trip plans based on network connectivity and routing rules. We run *od2trips* 1,000 times with randomized trip samples and retain only those OD pairs with at least one valid route. This filtering process ensures that all OD pairs included in the benchmark are feasible, and as a result, the actual number of OD pairs per network may be smaller than the theoretical maximum of $\tau(\tau - 1)$. Table 1 summarizes the number of TAZes and retained OD pairs for each benchmark network.

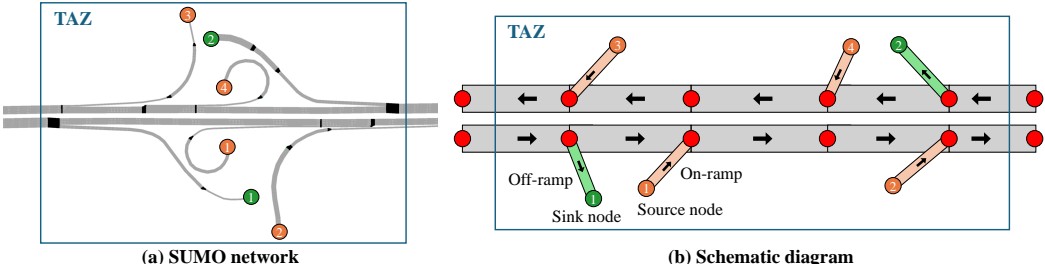

**(a) SUMO network**  **(b) Schematic diagram**

Figure 5: TAZ configuration example with source and sink nodes.

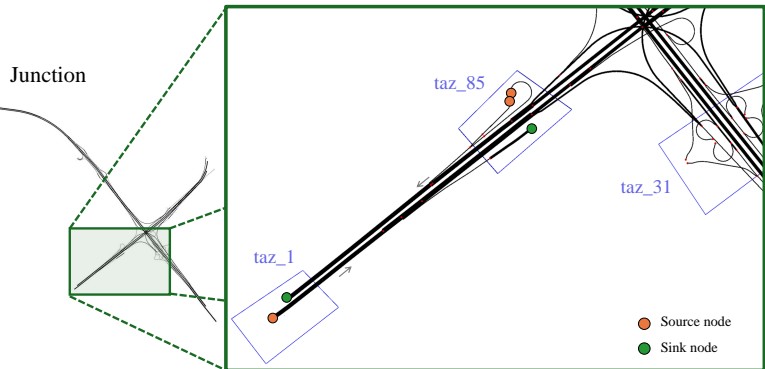

Figure 6: An example of infeasible OD pairs. Due to network directionality and scope, `taz_85` can only exchange demand with `taz_1`, making OD pairs involving other zones such as `taz_31` infeasible.

# B  Sensor filtering under data reliability and TAZ granularity constraints

To ensure the validity of our benchmark experiments, we filtered PeMS sensor data based on two main criteria: (1) physical consistency of traffic count patterns, and (2) structural ambiguity arising from TAZ definitions. This appendix describes our sensor exclusion methodology in detail.

## B.1  Sensor reliability filtering based on traffic count conservation principles

Traffic counts are expected to obey conservation principles on freeway networks. In particular, ML counts are expected to increase as traffic flows past on-ramps, due to merging vehicles, and decrease after off-ramps, due to diverging vehicles. To validate the reliability of sensor measurements, we checked whether these traffic count relationships held across time and space.

Let $M_{\text{up}}$ and $M_{\text{down}}$ denote the ML count observed upstream and downstream of a ramp, respectively. We applied the following consistency checks, as illustrated in Figure 7:

- At on-ramps, we expect: $M_{\text{up}} \leq M_{\text{down}}$
- At off-ramps, we expect: $M_{\text{up}} \geq M_{\text{down}}$

Count values that frequently violated these conditions—i.e., where the inequality was reversed—were flagged as inconsistent. Such violations suggest potential issues such as sensor misalignment, malfunction, or measurement noise. We removed all sensors associated with these links from the dataset to maintain physical plausibility in the OD calibration process. Note that the set of remaining sensors after this filtering step may vary depending on the date and time, as it is determined by the observed traffic counts.

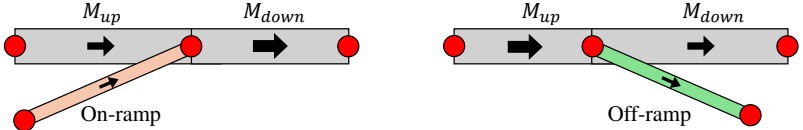

Figure 7: An illustration of traffic count conservation.

## B.2  Sensor filtering induced by TAZ granularity limitations

A second source of inconsistency stems from the coarse spatial resolution of TAZes. In our setting, each OD pair is defined at the TAZ level; however, multiple freeway access points (e.g., on-ramps and off-ramps) can exist within a single TAZ. This creates ambiguity when mapping OD flows to observed traffic counts, particularly when multiple access points from the same TAZ feed into different freeway links.

To address this issue, we identified and excluded sensors that lie: (1) between multiple on-ramps associated with a single origin TAZ (i.e., where source trips from the same TAZ may enter the freeway through different ramps), (2) between multiple off-ramps associated with a single destination TAZ (i.e., where sink trips may exit the freeway through different ramps). Such sensors may observe only a subset of the flows associated with a TAZ-level OD pair, leading to incomplete or misleading traffic count data. An illustrative case is shown in Figure 8. Removing these sensors ensures a more consistent correspondence between observed counts and counts from modeled OD, thereby improving the interpretability and fairness of the benchmark evaluation.

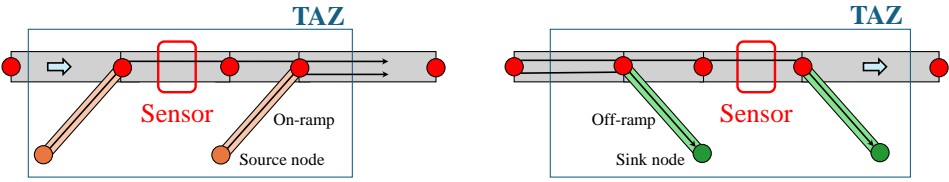

Figure 8: An example of sensor exclusion due to TAZ granularity issues.

## C  Implementation details

We summarize the configurations used for each optimization method evaluated in our main experiments.

### C.1  SPSA

At each iteration $k$, a random perturbation vector $\boldsymbol{\Delta}_k \in \{-1, +1\}^d$ is sampled, with each coordinate independently drawn with equal probability. The loss function is evaluated at two symmetric points $\mathbf{d}_k \pm c_k \boldsymbol{\Delta}_k$, and the gradient is approximated as

$$\hat{g}_k = \frac{f(\mathbf{d}_k + c_k \boldsymbol{\Delta}_k) - f(\mathbf{d}_k - c_k \boldsymbol{\Delta}_k)}{2c_k} \cdot \boldsymbol{\Delta}_k.$$

The next iterate is given by

$$\mathbf{d}_{k+1} = \mathbf{d}_k - a_k \hat{g}_k,$$

where $a_k = \frac{a}{(k+1+A)^\alpha}$ and $c_k = \frac{c}{(k+1)^\gamma}$ are decaying sequences controlling the step size and perturbation magnitude.

We use the recommended settings $\alpha = 0.602$ and $\gamma = 0.101$ [Spall, 2005]. The parameter $A$ is set to 10% of the total number of iterations, and $c = 0.1$. The parameter $a$ is computed as $a = 0.1 \times (1+A)^\alpha$, and rounded to two decimal places. All iterates $\mathbf{d}_k$ are clipped to the normalized domain $[0, 1]^d$, and the initial point is selected as the best solution from the initial design.

### C.2  Vanilla BO

We use a standard BO method with a Gaussian process surrogate and a Matérn $5/2$ kernel with automatic relevance determination (ARD). Lengthscales are constrained to $[0.005, 4.0]$, and a Gaussian likelihood is used with noise variance inferred in $[10^{-8}, 10^{-3}]$. Inputs are normalized to $[0, 1]^d$, and outputs are standardized. The acquisition function is q-Log Expected Improvement (qLogEI).

### C.3  SAASBO

We use the SAASBO method, implemented via BoTorch's `SaasFullyBayesianSingleTaskGP`. The model places a hierarchical horseshoe prior over the inverse lengthscales of a Matérn $5/2$ kernel to induce sparsity. Fully Bayesian inference is performed using the No-U-Turn Sampler (NUTS), with 32 warm-up steps, 16 posterior samples, and thinning interval of 16. The acquisition function is q-Expected Improvement (qEI). Inputs are normalized to $[0, 1]^d$, and outputs are standardized.

### C.4  TuRBO

We implement TuRBO with a single trust region. The region is initialized with length 0.8 and dynamically adjusted based on optimization performance: it is doubled after 3 consecutive improvements and halved after a failure count reaching $\lceil \max(4/b, \ d/b) \rceil$, where $b$ is the batch size. A restart is triggered if the region length drops below $0.5^7$, and the maximum length is capped at 1.6.

The region is anisotropic, shaped by the GP's learned lengthscales normalized by both their arithmetic and geometric means. The surrogate model uses a Matérn $5/2$ kernel with ARD, with lengthscales constrained to $[0.005, 4.0]$, and a Gaussian likelihood with noise bounded in $[10^{-8}, 10^{-3}]$. Candidates are generated using Sobol sequences with masked perturbations, where each dimension is perturbed with probability $\min(20/d, \ 1)$, and selection is performed via Thompson sampling. All inputs are normalized to $[0, 1]^d$, and outputs are standardized.

# D    Benchmark configurations

Table 3 summarizes the simulation and BO parameters used for each benchmark network. All networks use OD demand values bounded between 1 and 2,000 (or 2,500 for Simple Ramp). Simulation times, sensor observation times, OD generation time, and BO hyperparameters such as batch size and number of restarts are scaled according to network complexity.

Table 3: Simulation and BO configurations for the five benchmark networks.

| Network | Simulation time (sec) | Sensor time (sec) | OD time (sec) | Init. points | Epochs $T$ | Batch size | Num. restarts | Raw samples | Sample shape | Runs |
|---|---|---|---|---|---|---|---|---|---|---|
| Simple Ramp | 0–3600 | 0–3600 | 0–3300 | 10 | 50 | 2 | 8 | 128 | 64 | 10 |
| One-Way Corridor | 0–3900 | 300–3900 | 0–3600 | 20 | 100 | 3 | 16 | 256 | 64 | 10 |
| Junction | 0–3900 | 300–3900 | 0–3600 | 30 | 200 | 4 | 32 | 512 | 128 | 10 |
| Small Region | 0–4200 | 600–4200 | 0–3600 | 50 | 600 | 5 | 64 | 1024 | 128 | 3 |
| Full Region* | 0–4800 | 1200–4800 | 0–3600 | 20 | 5 | 2 | 32 | 512 | 128 | 1 |

*Reduced settings were used for Full Region due to computational constraints.

*Column descriptions:*

- Simulation time (sec): Total duration of the traffic simulation.
- Sensor time (sec): Time window during which traffic sensors collect observations.
- OD time (sec): Time interval over which OD demand is generated.
- Init. points: Number of initial samples before optimization model starts.
- Epochs $T$: Total number of optimization model iterations.
- Batch size: Number of candidate evaluations per epoch.
- Num. restarts: Number of restarts used for optimizing the acquisition function.
- Raw samples: Number of raw samples used to seed acquisition optimization.
- Sample shape: Number of posterior samples used to estimate the acquisition function via Monte Carlo. Relevant only for Vanilla BO.
- Runs: Number of independent optimization runs, each using a different random seed.

# E    Detailed experimental results

Figure 9 illustrates the alignment between simulated and GT traffic counts for the four networks other than the Small Region discussed in the main text. Black dots represent the mean values, and the lengths of the error bars indicate one standard deviation. As in the Small Region case, SPSA consistently demonstrates the poorest performance across all networks. In the Simple Ramp network (see Figure 9a), the simulated link traffic counts closely match the GT values for all BO models. For the One-Way Corridor (see Figure 9b), all BO models exhibit nearly linear correspondence. In the Junction network (see Figure 9c), all BO models show near-linear alignment, except for a few points slightly deviating from the diagonal. Vanilla BO and TuRBO achieve better overall performance, while SAASBO shows relatively longer error bars, indicating greater variability.

Table 4: NRMSE range and average for each model on the Full Region, excluding the initial solution. Results are based on the minimum loss observed over 5 epochs.

| Value | Model | | | |
|---|---|---|---|---|
| | SPSA | Vanilla BO | SAASBO | TuRBO |
| Minimum | 0.782 | 0.771 | 0.775 | **0.769** |
| Maximum | 0.792 | 0.793 | 0.791 | **0.787** |
| Average | 0.787 | 0.784 | 0.784 | **0.779** |

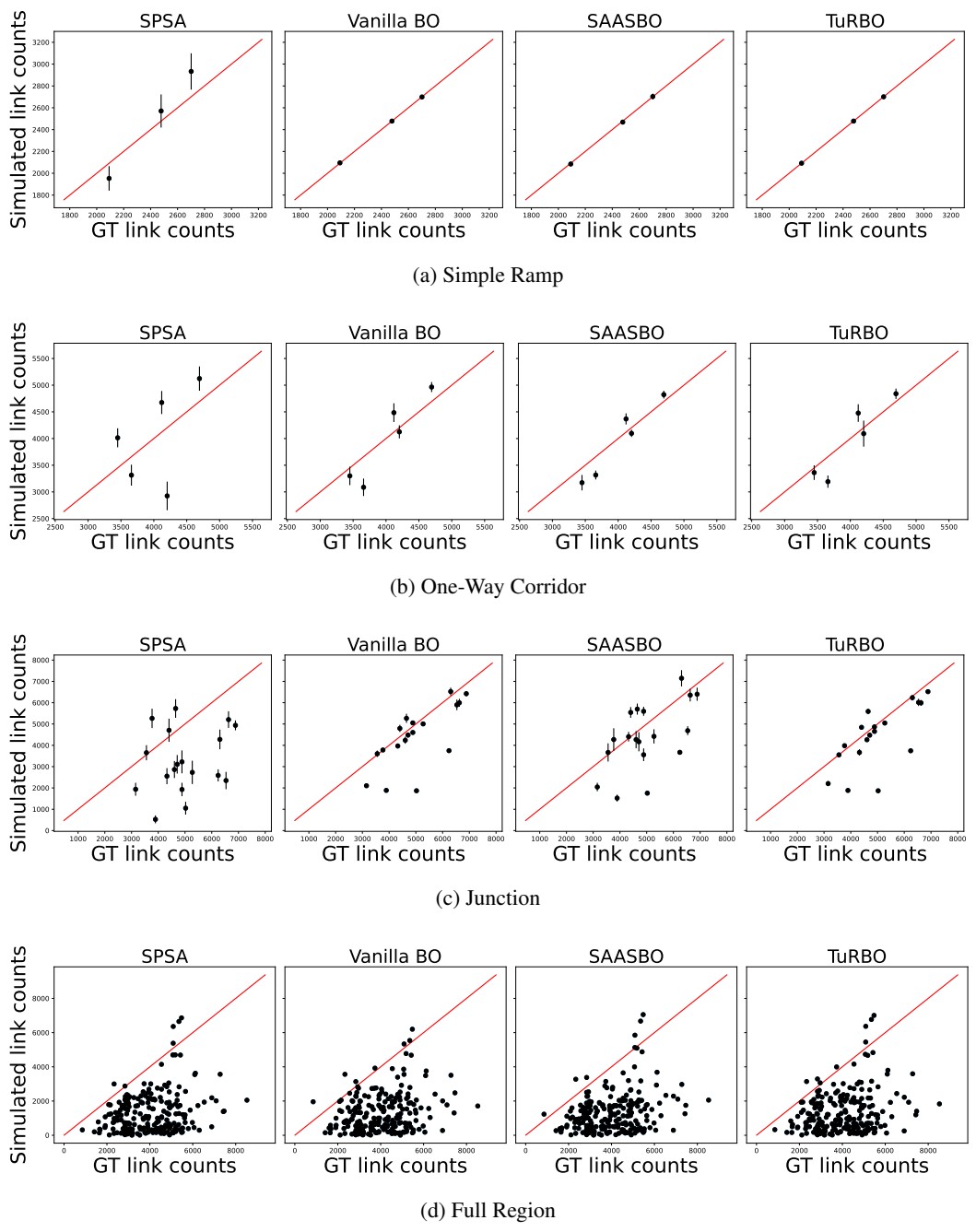

(a) Simple Ramp

(b) One-Way Corridor

(c) Junction

(d) Full Region

Figure 9: Fit to GT link traffic counts for (a) Simple Ramp, (b) One-Way Corridor, (C) Junction, and (d) Full Region.

Despite an average NRMSE of 0.783 and a minimum of 0.765 among the 20 initial samples, the optimization process in the Full Region setting failed to achieve any reduction beyond the initial solutions (see Table 4). These results exclude the initial solution and reflect performance after the first epoch. While TuRBO produced the lowest NRMSE among the models tested, the differences across models were marginal and not statistically meaningful.

Inspection of Figure 9d reveals that, in many cases, the simulated link counts were consistently lower than the GT values. One possible explanation is that large OD demand values may have led to traffic congestion in the simulation, preventing vehicles from reaching the sensors within the active sensor collection period. To address this issue, we suggest refining key simulation and demand-related

parameters. This includes calibrating the upper bound of OD demand values, which is currently set to 2,000, to avoid excessive congestion, and adjusting simulation settings such as simulation time, sensor time, and OD time (see Table 3) to ensure consistency between demand generation and measurement.

## F    Effect of excluding unobservable OD pairs on optimization performance

To evaluate the impact of including OD pairs that do not contribute to sensor measurements, we conduct an experiment comparing optimization performance with and without such unobservable OD pairs. These pairs correspond to OD flows that are not routed through any links with active sensors and therefore do not directly influence any measured traffic.

We conduct this analysis on a different time window than the main experiments. Accordingly, the OD upper bound was scaled down to 1,138 in proportion to the total sensor-measured traffic counts, which were lower than in the main setting. Out of the 21 OD pairs in the One-Way Corridor network, 4 were identified as unobservable. These correspond to OD flows from `taz_0` to `taz_60`, from `taz_61` to `taz_62`, from `taz_63` to `taz_64`, and from `taz_64` to `taz_1`. These OD pairs are visualized in Figure 10.

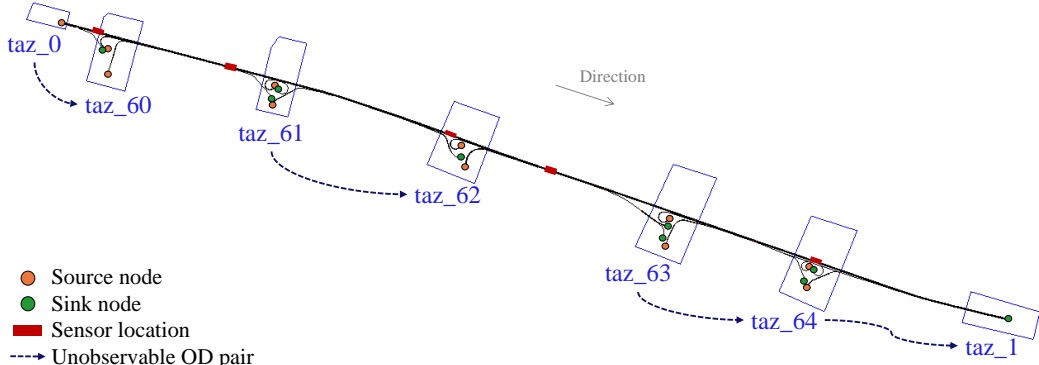

Figure 10: Visualization of the 4 unobservable OD pairs in the One-Way Corridor network. These OD flows do not pass through any sensor-instrumented link and were excluded in the filtered setting.

We compare two settings: one using the full set of 21 OD pairs, and another using a filtered set of 17 OD pairs with unobservable pairs excluded. For each setting, we ran 10 independent experiments using the same initialization, enabling a controlled and fair comparison of optimization performance. Notably, even though the initial candidate pool is shared, excluding unobservable OD pairs can alter simulation dynamics, since these flows, while not directly observed, may still influence congestion dynamics that affect sensor readings indirectly. As a result, initial NRMSE values may differ between the two settings.

Table 5 and Figure 11 summarize the optimization outcomes for each model comparing the inclusion and exclusion of unobservable OD pairs. The filtered setting, despite starting with a higher initial NRMSE on average, consistently achieved greater improvement across all models. While the best NRMSE of SPSA slightly increased when unobservable OD pairs were excluded, all Bayesian optimization methods exhibited improved estimation performance. In particular, TuRBO and Vanilla BO showed substantial reductions in best NRMSE, with TuRBO achieving the most dramatic improvement. SAASBO also benefited from the exclusion, though the magnitude of improvement was more modest. These results suggest that removing unobservable OD variables not only improves sample efficiency but can also enhance final estimation quality, particularly for surrogate-based methods whose performance can deteriorate in the presence of irrelevant input dimensions. This highlights the value of aligning the OD parameter space with sensor-observable flows, which can lead to more robust and efficient optimization outcomes. At the same time, the presence of unobservable OD pairs also underscores the need for adaptive feature selection strategies that can selectively filter such variables during optimization.

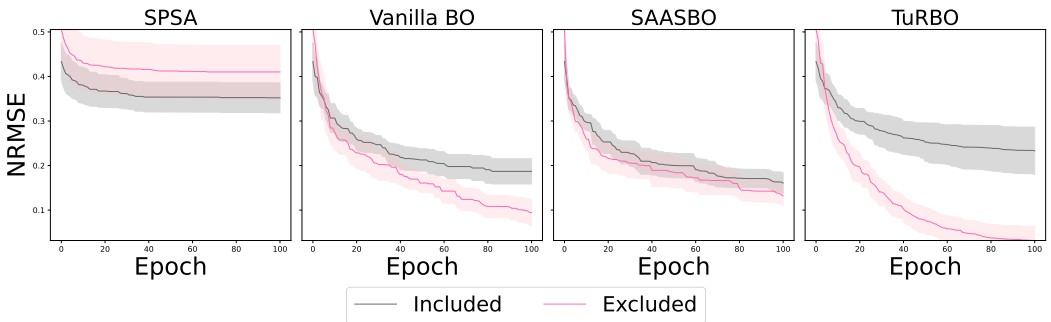

Figure 11: Comparison of optimization performance including and excluding unobservable OD pairs. Solid lines represent the mean, with error bars denoting one standard deviation over 10 runs.

Table 5: Average best NRMSE values for each model on the One-Way Corridor network, comparing settings with and without unobservable OD pairs. Values in parentheses indicate the average percentage improvement. Both metrics are averaged across 10 independent runs.

| Unobservable OD pairs | Initial solution | | Model | | | |
|---|---|---|---|---|---|---|
| | Avg | Min | SPSA | Vanilla BO | SAASBO | TuRBO |
| Included | 0.722 | 0.434 | **0.352** (18.03%) | 0.187 (56.72%) | 0.161 (62.50%) | 0.233 (49.04%) |
| Excluded | 0.815 | 0.505 | 0.410 **(18.64%)** | **0.094 (80.62%)** | **0.132 (71.54%)** | **0.032 (93.81%)** |

## G    Effect of kernel selection on optimization performance

An additional experiment is conducted to analyze how the choice of GP kernel affects BO performance. The experimental setup is identical to that used for the main results in Table 2, with all conditions kept the same except for the kernel selection. Three GP kernels, Matérn 3/2, Matérn 5/2, and RBF, are evaluated across four network types and three BO variants (Vanilla BO, SAASBO, and TuRBO). SAASBO is implemented using the SaasFullyBayesianSingleTaskGP[5], which employs a Matérn 5/2 by default, so this variant is tested only with that kernel. The Matérn 5/2 results are consistent with those reported in Table 2, except for the Small Region case, where a smaller number of runs is used in this supplementary experiment. The additional results for Matérn 3/2 and RBF kernels are included here for comparison (Table 6). Overall, the results indicate that although the Matérn 3/2 kernel shows a slight performance advantage in several cases, the differences among kernels remain small.

Table 6: Average best NRMSE across four network types and three BO models under different GP kernels. The number of independent runs for each network is indicated in the Runs column.

| Network | Runs | Vanilla BO | | | SAASBO | TuRBO | | |
|---|---|---|---|---|---|---|---|---|
| | | Matérn 3/2 | Matérn 5/2 | RBF | Matérn 5/2 | Matérn 3/2 | Matérn 5/2 | RBF |
| Simple Ramp | 10 | 0.0011 | 0.0033 | 0.0074 | 0.0166 | **0.0002** | 0.0007 | 0.0012 |
| One-Way Corridor | 10 | 0.0981 | 0.1055 | 0.1121 | **0.0703** | 0.0840 | 0.0898 | 0.0934 |
| Junction | 10 | 0.2344 | **0.2326** | 0.2362 | 0.3017 | 0.2376 | 0.2337 | 0.2354 |
| Small Region | 1 | 0.3412 | 0.5006 | 0.4558 | 0.5413 | 0.2074 | 0.1483 | **0.1260** |

## H    Verification of the high-dimensionality of BO4Mob

To verify that the BO4Mob problems exhibit genuinely high-dimensional characteristics, we analyze the influence of each input variable (i.e., each OD pair) on the objective function. Following the

---

[5]https://botorch.readthedocs.io/en/latest_modules/botorch/models/fully_bayesian.html#SaasFullyBayesianSingleTaskGP

approach proposed in Papenmeier et al. [2025], we identify *dominant* and *secondary* variables based on input sensitivity. Table 7 summarizes the average number of dominant and secondary variables for each network instance. Unlike the commonly used high-dimensional benchmarks such as LassoBench [Šehić et al., 2022] and MOPTA08 [Eriksson and Jankowiak, 2021], where many variables are found to have negligible effects, our results indicate that BO4Mob problems involve a much higher proportion of influential variables. For example, in the Small Region network with 151 OD variables, over 95% (144.3 on average) are identified as dominant. This suggests that BO4Mob captures truly high-dimensional and complex optimization settings, where most input features significantly affect the objective. Overall, the results confirm that BO4Mob encompasses complex and high-dimensional optimization settings suitable for evaluating the performance of black-box optimization algorithms.

## I  Configurability and extensibility of the benchmark framework

The BO4Mob benchmark offers high configurability, allowing users to easily set up diverse experimental conditions through flexible command-line arguments and a modular experiment design. Users can specify the network type, optimization model, GP kernel, simulation date and time window, and evaluation measure. As illustrated in Appendix G, the framework supports several representative GP kernels, including Matérn 1.5, Matérn 2.5, and RBF. While the main experiments in the paper used link counts as the evaluation measure, the benchmark can also be configured to use average speed for performance assessment. This configurability enables users to explore a broad range of experimental settings without code modification, ensuring fair comparisons across models.

The BO4Mob framework is designed to support extensibility through clearly documented implementation guides. The released codebase provides instructions on how the network and sensor data were prepared, allowing users to apply the framework to additional networks by following the same preprocessing steps with their own data. It also includes guidelines for integrating newly developed optimization methods or GP kernels into the existing pipeline. While the benchmark currently implements Matérn 1.5, Matérn 2.5, and RBF kernels, the modular structure enables users to substitute or add domain-specific alternatives as needed. This extensibility allows BO4Mob to serve as a foundation for future research exploring novel BO methods and applications in transportation modeling.

## J  SPSA as a non-BO baseline

SPSA is adopted as a non-BO baseline because it is widely used for OD estimation in both academic studies [Balakrishna et al., 2007, Vaze et al., 2009, Cipriani et al., 2011, Ben-Akiva et al., 2012, Ros-Roca et al., 2021] and real-world applications [Galgano et al., 2021, Talas et al., 2021, Ban et al., 2022], including projects funded by the USDOT. Traditional approaches in the transportation literature often rely on SPSA and its variants, which approximate gradients under noisy simulation settings. The method remains attractive due to its algorithmic simplicity, low evaluation cost, and suitability for simulation-based optimization. However, SPSA's limitations in high-dimensional and noisy environments are well documented [Antoniou et al., 2015, Tympakianaki et al., 2015, Qurashi et al., 2019], including unstable gradient estimates and the absence of explicit structural modeling. Although advanced variants such as W–SPSA [Antoniou et al., 2015], c–SPSA [Tympakianaki et al., 2015], and PC–SPSA [Qurashi et al., 2019] address some of these issues, they typically require prior parameter tuning, access to historical data, or manual design of weight matrices, and still face challenges when applied at scale.

Table 7: Average counts of dominant and secondary OD variables across network types.

| Network | # Dominant | # Secondary |
|---|---|---|
| Simple Ramp | 3.0 | 0.0 |
| One-Way Corridor | 19.3 | 1.7 |
| Junction | 35.6 | 8.4 |
| Small Region | 144.3 | 6.7 |

# K Comparison of evaluation measures: link count vs. average speed

To examine the effect of the evaluation measure on optimization identifiability, we conduct a simple experiment on the Simple Ramp configuration, which contains only three deterministic OD pairs. All other settings are identical, and only the evaluation measure (either link count or average speed) is varied. As summarized in Table 8, the count-based experiment rapidly converges from an initial best NRMSE of 0.1408 to 0.0043, successfully reconstructing the GT OD demands. In contrast, when average speed is used, the optimization shows almost no improvement in NRMSE (remaining around 0.023 throughout) and yields unrealistic OD estimates, where the first two OD pairs reach the minimum feasible value of 1 while the third saturates at the upper bound of 2500. These results suggest that, while the framework supports both count- and speed-based evaluation options, the use of link counts is generally more informative and is therefore recommended for OD estimation within BO4Mob.

Table 8: Comparison of OD estimation results using different evaluation measures.

| Type | 1st OD | 2nd OD | 3rd OD | Initial best NRMSE | Final best NRMSE |
|------|--------|--------|--------|--------------------|------------------|
| GT | 2092 | 609 | 386 | – | – |
| Count | 2074.9 | 629.5 | 408.3 | 0.1408 | 0.0043 |
| Average speed | 1 | 1 | 2500 | 0.0239 | 0.0233 |

# L Limitations

A limitation of our benchmark is that it includes only five predefined subnetworks from a single region (the San Francisco Bay Area), including the Full Region. Users cannot flexibly extract arbitrary subnetworks, which limits adaptability to custom or unseen scenarios beyond the provided configurations.

Another limitation is related to the computational cost of simulating the Full Region. Due to its high memory usage and long simulation time, we were only able to run a limited number of epochs. Although increasing the batch size would help explore a wider range of solutions, it also significantly increases memory consumption. In practice, we set the batch size to 2 to fit within resource constraints, but this led to incomplete results within the allocated runtime. This suggests that running optimization over the Full Region requires sufficient RAM and prior analysis of SUMO's peak memory usage to choose an appropriate batch size.

# M License

This benchmark incorporates external resources that are redistributed or referenced in accordance with their original licenses:

- **Road network files:** Provided by ETH Zurich and licensed under the Creative Commons Attribution-NonCommercial 4.0 International License (CC BY-NC 4.0).

- **Traffic sensor data (PeMS):** Traffic count data is sourced from PeMS. Use of this data is subject to the Caltrans Conditions of Use: `https://dot.ca.gov/conditions-of-use`.

- **Traffic simulation (SUMO):** We use the open-source SUMO traffic simulator (version 1.12), which is distributed under the Eclipse Public License 2.0 (EPL-2.0).

- **Bayesian optimization library (BoTorch):** Our implementation uses BoTorch, which is licensed under the MIT License.

# N  Datasheet

We adopt the datasheet framework introduced by Gebru et al. [2021] to document our benchmark datasets.

## N.1  Motivation

**For what purpose was the dataset created? Was there a specific task in mind? Was there a specific gap that needed to be filled? Please provide a description.**

The datasets in this benchmark were constructed to support the evaluation and comparison of optimization-based methods for OD demand estimation in transportation networks. By simulating five predefined networks with known GT and sensor configurations, the benchmark enables controlled, reproducible experiments. In particular, it is designed to facilitate the development and analysis of BO techniques by providing a structured setting.

**Who created the dataset (for example, which team, research group) and on behalf of which entity (for example, company, institution, organization)?**

This dataset was created by Seunghee Ryu, Donghoon Kwon, Seongjin Choi, Aryan Deshwal, Seungmo Kang, Carolina Osorio, who are researchers affiliated with University of Minnesota, Korea University, HEC Montréal.

**Who funded the creation of the dataset? If there is an associated grant, please provide the name of the grantor and the grant name and number.**

Seunghee Ryu and Donghoon Kwon are affiliated with the Department of Civil, Environmental and Architectural Engineering, Korea University, South Korea, and are supported by the Basic Science Research Program through the National Research Foundation of Korea (NRF), funded by the Ministry of Education, South Korea (RS-2020-NR049594), and by the BK21 FOUR (Brain Korea 21 Four) Project; Support Program for Outstanding Graduate Students' International Joint Training. Seongjin Choi is supported by the Department of Civil, Environmental and Geo-Engineering and Center for Transportation Studies at the University of Minnesota. Seungmo Kang is supported by the Basic Science Research Program through the NRF, funded by the Ministry of Education, South Korea (RS-2020-NR049594).

## N.2  Composition

**What do the instances that comprise the dataset represent (for example, documents, photos, people, countries)? Are there multiple types of instances (for example, movies, users, and ratings; people and interactions between them; nodes and edges)? Please provide a description.**

The dataset consists of two types of instances: (1) network-related files required to run SUMO traffic simulations, including XML files for the network structure, OD, TAZ, and additional simulation parameters, as well as a CSV file describing the routes between TAZes; (2) link traffic count and average speed data originally obtained from the Caltrans PeMS. This data was post-processed by the researchers to match each sensor reading to a corresponding link in the SUMO simulation network, resulting in link-level traffic count information.

**How many instances are there in total (of each type, if appropriate)?**

The entire freeway network includes 1,977 nodes and 2,173 links, from which five subnetworks of varying sizes were derived, including the Full Region network. For traffic sensing, 4,080 sensors were available on October 22, 2022; 219 sensors were selected based on predefined filtering criteria (see Appendix B). To support extended analysis and benchmarking, the dataset includes traffic count data collected over a 14-day period (October 8–21, 2022), covering three daily time windows: 6:00–7:00 a.m., 8:00–9:00 a.m., and 5:00–6:00 p.m.

**Does the dataset contain all possible instances or is it a sample (not necessarily random) of instances from a larger set? If the dataset is a sample, then what is the larger set? Is the sample representative of the larger set (e.g., geographic coverage)? If so, please describe how this representativeness was validated/verified.**

The dataset represents a curated subset extracted from a larger real-world dataset. The networks are derived from freeway segments in the San Jose area, and the sensor data is selected from within these network boundaries. Specifically, we use only the ML detectors as defined in the original data source.

**What data does each instance consist of? "Raw" data (for example, unprocessed text or images) or features? In either case, please provide a description.**

The dataset includes lightly processed raw inputs. For the networks, the Full Region corresponds to a subset of the San Francisco Bay Area freeway network around San Jose, extracted from a larger SUMO network. Four additional subnetworks were constructed by selecting smaller subregions. For the sensor data, raw ML sensor locations (latitude and longitude) from PeMS were projected onto the network and matched to specific links. While the original data sources are raw, these instances have been curated to support simulation and analysis.

**Is there a label or target associated with each instance? If so, please provide a description.**

Yes. Each instance includes link traffic count (or average speed) values that serve as target outputs in the optimization process. These values represent the observed traffic count (or average speed) on each link and are used to evaluate the quality of OD demand estimates during optimization.

**Is any information missing from individual instances? If so, please provide a description, explaining why this information is missing (for example, because it was unavailable).**

No.

**Are relationships between individual instances made explicit (for example, users' movie ratings, social network links)? If so, please describe how these relationships are made explicit.**

Yes, relationships between sensor data and the network are made explicit. Each traffic sensor is associated with a specific link in the road network, allowing observed traffic counts (or average speed) to be directly mapped to network topology. These link-level associations are predefined and included as part of the dataset.

**Are there recommended data splits (for example, training, development/validation, testing)? If so, please provide a description of these splits, explaining the rationale behind them.**

No predefined data splits are provided. The dataset is intended to be used as a benchmark for evaluating black-box optimization algorithms on OD demand estimation tasks.

**Are there any errors, sources of noise, or redundancies in the dataset? If so, please provide a description.**

Yes, the raw sensor data from PeMS may include noise or inconsistent measurements. However, filtering criteria were applied to exclude unreliable sensors.

**Is the dataset self-contained, or does it link to or otherwise rely on external resources (for example, websites, tweets, other datasets)?**

Yes, the dataset is self-contained.

**Does the dataset contain data that might be considered confidential (for example, data that is protected by legal privilege or by doctor-patient confidentiality, data that includes the content of individuals' non-public communications)? If so, please provide a description.**

No, all data is derived from publicly available sources.

**Does the dataset contain data that, if viewed directly, might be offensive, insulting, threatening, or might otherwise cause anxiety? If so, please describe why.**

No, the dataset contains no offensive or disturbing content.

### N.3  Collection process

**How was the data associated with each instance acquired? Was the data directly observable (for example, raw text, movie ratings), reported by subjects (for example, survey responses), or indirectly inferred/derived from other data (for example, part-of-speech tags, model-based guesses for age or language)?**

Each instance includes a combination of directly observed and derived data. The sensor data originates from Caltrans PeMS, and the network files were adapted from Supplementary Note 2 of Ambühl et al. [2023].

**What mechanisms or procedures were used to collect the data (for example, hardware apparatuses or sensors, manual human curation, software programs, software APIs)?**

Sensor lists, traffic count, and average speed data were downloaded from the Caltrans PeMS website.[6] The network files were obtained from a previous study [Ambühl et al., 2023] and further processed. The matching between sensors and network links was performed manually by the researchers.

**If the dataset is a sample from a larger set, what was the sampling strategy (for example, deterministic, probabilistic with specific sampling probabilities)?**

The dataset is a deterministic sample. Subnetworks were selected based on geographic coverage and scalability, and sensors were filtered using predefined criteria such as location type (ML) and data reliability (see Appendix B).

**Who was involved in the data collection process (for example, students, crowdworkers, contractors) and how were they compensated (for example, how much were crowdworkers paid)?**

The data collection and processing were performed by the researchers involved in this study. No external contributors were involved or compensated.

**Over what timeframe was the data collected? Does this timeframe match the creation timeframe of the data associated with the instances (e.g., recent crawl of old news articles)? If not, please describe the timeframe in which the data associated with the instances was created.**

The sensor data was collected over two weeks from October 8 to October 21, 2022. For each day, traffic counts and average speed were extracted for three time windows: 6:00–7:00 a.m., 8:00–9:00 a.m., and 5:00–6:00 p.m. The metadata used to associate sensors with specific network links is based on the configuration as of October 22, 2022. The network files were sourced directly from those provided in Ambühl et al. [2023], which were released as part of the supplementary material on December 5, 2022.

**Were any ethical review processes conducted (for example, by an institutional review board)?**

No ethical review was required, as the dataset does not involve human subjects or sensitive information.

### N.4   Preprocessing/cleaning/labeling

**Was any preprocessing/cleaning/labeling of the data done (for example, discretization or bucketing, tokenization, part-of-speech tagging, SIFT feature extraction, removal of instances, processing of missing values)?**

Yes. The network is a cropped subset of the San Francisco Bay Area freeway network, adapted from the traffic simulation dataset provided by Ambühl et al. [2023]. Traffic count and average speed data from the Caltrans PeMS system were aggregated to 5-minute intervals. Only sensors labeled as ML were used. Each sensor's location was matched to the nearest freeway link with the same direction of traffic (as indicated by the "dir" column) to ensure accurate alignment with the network structure.

**Was the "raw" data saved in addition to the preprocessed/cleaned/labeled data (for example, to support unanticipated future uses)?**

No, the raw data is not included in the released dataset. However, the original sensor data can be downloaded from the Caltrans PeMS website (https://pems.dot.ca.gov/), and the original network files are available from the ETH Zurich research repository (https://www.research-collection.ethz.ch/handle/20.500.11850/584669).

**Is the software that was used to preprocess/clean/label the data available? If so, please provide a link or other access point.**

No. The code used for preprocessing is not included in the released dataset.

---

[6] `https://pems.dot.ca.gov/`

### N.5 Uses

**Has the dataset been used for any tasks already?**

Yes. Parts of the network have been used in prior work on OD estimation frameworks, but different subregions were used, and the PeMS data was from a different date than ours.

**Is there a repository that links to any or all papers or systems that use the dataset?**

Yes. The dataset and code are available at [https://github.com/UMN-Choi-Lab/BO4Mob], and a dedicated repository listing the dataset is maintained at [https://github.com/UMN-Choi-Lab/BO4Mob_data].

**What (other) tasks could the dataset be used for?**

While this dataset is used for OD estimation within traffic simulation in this study, it could also support other traffic simulation research tasks.

**Is there anything about the composition of the dataset or the way it was collected and preprocessed/cleaned/labeled that might impact future uses?**

The dataset has been preprocessed and is ready for direct use. It includes five predefined network levels, which support multi-scale analysis but do not allow user-defined regions. Additionally, sensor readings may contain noise due to equipment or environmental factors.

**Are there tasks for which the dataset should not be used? If so, please provide a description.**

No, there are no known tasks for which the dataset should not be used.

### N.6 Distribution

**Will the dataset be distributed to third parties outside of the entity (for example, company, institution, organization) on behalf of which the dataset was created?**

Yes. The dataset has been publicly released and is available to third parties via a GitHub repository.

**How will the dataset be distributed (for example, tarball on website, API, GitHub)? Does the dataset have a digital object identifier (DOI)?**

The code and dataset used in our benchmark study have been made publicly available via a public GitHub repository at [https://github.com/UMN-Choi-Lab/BO4Mob]. A DOI is not provided.

**When will the dataset be distributed?**

On May 15, 2025.

**Will the dataset be distributed under a copyright or other intellectual property (IP) license, and/or under applicable terms of use (ToU)? If so, please describe this license and/or ToU, and provide a link or other access point to.**

Yes. The benchmark dataset is released under the CC BY 4.0 International License (`https://creativecommons.org/licenses/by/4.0`). The code implementation is released under the MIT License (`https://opensource.org/license/MIT`).

**Have any third parties imposed IP-based or other restrictions on the data associated with the instances? If so, please describe these restrictions, and provide a link or other access point to, or otherwise reproduce, any relevant licensing terms, as well as any fees associated with these restrictions.**

No. There are no IP-based restrictions known, but users should refer to the original sources—Caltrans PeMS (`https://pems.dot.ca.gov/`) and the ETH Zurich research repository (`https://www.research-collection.ethz.ch/handle/20.500.11850/584669`)—for their respective terms of use.

**Do any export controls or other regulatory restrictions apply to the dataset or to individual instances? If so, please describe these restrictions, and provide a link or other access point to, or otherwise reproduce, any supporting documentation.**

No.

### N.7 Maintenance

**Who will be supporting/hosting/maintaining the dataset?**

The dataset will be maintained by the authors of this paper.

**How can the owner/curator/manager of the dataset be contacted (for example, email address)?**

Please contact the following email address: chois@umn.edu

**Is there an erratum? If so, please provide a link or other access point.**

Any future corrections or updates will be documented in the GitHub repository.

**Will the dataset be updated (e.g., to correct labeling errors, add new instances, delete instances)? If so, please describe how often, by whom, and how updates will be communicated to dataset consumers (e.g., mailing list, GitHub)?**

Yes. Any updates will be managed by the authors and documented in the GitHub repository.

**If the dataset relates to people, are there applicable limits on the retention of the data associated with the instances (e.g., were the individuals in question told that their data would be retained for a fixed period of time and then deleted)? If so, please describe these limits and explain how they will be enforced.**

The dataset does not contain any data relating to people.

**Will older versions of the dataset continue to be supported/hosted/maintained? If so, please describe how. If not, please describe how its obsolescence will be communicated to dataset consumers.**

Yes. If updates are made, older versions of the dataset will remain available and accessible through the GitHub repository.

**If others want to extend/augment/build on/contribute to the dataset, is there a mechanism for them to do so? If so, please provide a description. Will these contributions be validated/verified? If so, please describe how. If not, why not? Is there a process for communicating/distributing these contributions to dataset consumers? If so, please provide a description.**

Yes. Others can contribute by opening a GitHub issue or by contacting the corresponding author by email.

