# OpenReview forum: "BO4Mob: Bayesian Optimization Benchmarks for High-Dimensional Urban Mobility Problem"
_NeurIPS.cc/2025/Datasets_and_Benchmarks_Track — NeurIPS 2025 Datasets and Benchmarks Track poster_

### Official Review · Reviewer_wSpt · 2025-06-30

**Rating:** 5
**Confidence:** 4

**Summary:**

This paper introduces BO4Mob, a new open-source benchmarking framework designed to address the highly challenging problem of estimating high-dimensional origin-destination (OD) demand in urban transportation. This problem requires inferring travel demand across the entire transportation network based on limited sensor data, with the challenges lying in its high dimensionality, high cost of single-run traffic simulations, and the random and non-differentiable nature of the process. The BO4Mob framework includes five scenarios based on the real road network of San Jose, California, with increasing complexity, reaching a maximum dimension of 10,100. It utilizes the high-fidelity SUMO traffic simulator and real PeMS sensor data to provide a realistic and reproducible testing platform for evaluating and developing advanced Bayesian optimization (BO) algorithms.

**Dataset Code Accessibility:**

Yes

**Dataset Code Comments:**

All data and code are publicly available.

**Ethical Comments:**

This is a traffic simulation optimization platform. All data is obtained from open sources, and there are no ethical concerns.

**Ethical Considerations:**

No, there are no or only very minor ethics concerns

**Final Justification:**

The BO4Mob dataset proposed in this paper incorporates a traffic simulator and Bayesian optimization algorithms , which can be used to solve the problem of estimating travel demand with limited data. So, I recommend to accept this paper.

**Limitations Weaknesses:**

1. The platform provided in this study consists of five predefined, fixed networks. While this facilitates standardized comparisons, it also limits the flexibility and broad applicability of the evaluation.
2. The paper positions the “complete region” scenario as the ultimate test of algorithm scalability, but the experimental analysis of this scenario is limited. This scenario was only run once independently and iterated five times, which is insufficient compared to other scenarios.
3. The only non-BO baseline in the study was SPSA. As can be seen from Figure 3 and Table 2, SPSA performed very poorly in all scenarios and had almost no optimization capability. This seems to lead readers to believe that non-BO methods are not applicable to this task.

**Strengths Contributions:**

1. This paper bridges the gap between advanced Bayesian optimization (BO) algorithm research and real-world traffic engineering applications, addressing the issue of the lack of a standardized testing platform.
2. The proposed framework integrates real-world road network data, combines high-fidelity open-source traffic simulators (SUMO) and real sensor traffic data (PeMS), ensuring the realism and complexity of the test scenarios.
3. The proposed framework is scalable and provides five traffic network scenarios ranging from simple to complex, enabling systematic evaluation of the performance and scalability of optimization algorithms at different scales and levels of difficulty.
4. The proposed framework provides a challenging “testing ground” for the research community, promoting progress in related fields, like Bayesian optimization and traffic simulation.

---

> ### Author Rebuttal · Authors · 2025-07-30
>
> ---
>
> We sincerely thank you for your constructive and thoughtful feedback. Below, we address concerns raised in your review.
>
> ---
>
> ## Reviewer Comment 1: Limited Flexibility Due to Predefined Fixed Networks
>
> > *"The platform provided in this study consists of five predefined, fixed networks. While this facilitates standardized comparisons, it also limits the flexibility and broad applicability of the evaluation."*
>
> **Response:**
> We agree that using a fixed set of networks introduces some limitations. However, as noted by you and other reviewers, the realism of the scenarios is one of the benchmark’s key strengths. Most transportation research papers rely on a single case study due to the time-intensive and challenging nature of network setup, making our inclusion of five real networks particularly valuable for comprehensive evaluation. This realism makes the benchmark both **challenging and practically relevant**, and it also explains the significant engineering effort required to develop it.
> While our benchmark currently includes five carefully constructed subnetworks from the San Francisco Bay Area, these networks were selected and processed through a time-intensive curation process that ensures their meaningfulness and quality. Specifically, the  thorough curation process for these networks involved three critical components that ensure their high quality and realism:
> - **Zone-by-zone vehicle generation mapping**: Each network required manual defining traffic generation zones to accurately reflect real-world traffic patterns.
> - **Source-sink point calibration**: We established precise criteria for vehicle generation (source) and termination (sink) points within each zone to ensure realistic traffic flow dynamics.
> - **Multi-scale hierarchical construction**: This labor-intensive process was applied across multiple network scales—from individual ramps to corridors (connected ramps), junctions (corridor intersections), small regions (connected junctions), and a full region—creating a comprehensive hierarchical evaluation framework.
>
>
> This **meticulous curation process** ensures that each network provides meaningful and realistic evaluation scenarios, which is why we believe the current five networks offer substantial value despite the numerical limitation. The infrastructure is designed to support scalability and extension.
> Specifically:
>
> - Users can build new SUMO networks using open-source tools such as `netconvert` with data from OpenStreetMap data.
> - We will include a new section in the Appendix (titled *“How to Add New Networks”*) that documents:
>   - Configuration file structure,
>   - Steps to integrate custom networks,
>   - Relevant tools and SUMO documentation,
>   - Guidelines for mapping sensor metadata to simulation links.
>  We believe the provided guidelines will help researchers apply BO4Mob to their own scenarios, while maintaining the rigor and realism of the original design.
>
> ---
>
> ## Reviewer Comment 2: Limited Experimental Analysis of the Full Region Scenario
>
> > *"The paper positions the “complete region” scenario as the ultimate test of algorithm scalability, but the experimental analysis of this scenario is limited. …"*
>
> **Response:**
> Thank you for this important observation. We agree that the current analysis of the Full Region scenario is limited due to practical constraints. As noted in Appendix E, the optimization process failed to improve upon the initial samples, and in Appendix G we explain that:
>
> - Each simulation run on the Full Region takes **over 11 hours** (even without optimization),
> - Memory limits required **batch sizes as low as 2**, which led to very slow convergence,
> - We were constrained to running **just five optimization epochs**.
>
> **As an academic lab, these constraints are difficult to overcome**. However, we believe the fact that the pipeline runs at all on such a large-scale, simulation-driven setting is a meaningful result. The Full Region scenario is intentionally extreme, and its inclusion highlights an important open challenge: **current BO methods struggle under such ultra-high-dimensional, resource-intensive conditions.**
>
> Rather than offering a complete solution, we aim to **expose this limitation** and provide a foundation for future research on scalable BO. We believe making this scenario available, despite its difficulty, is valuable for the community.
>
> ---
>
> ## Reviewer Comment 3: Choice and Performance of the Non-BO Baseline (SPSA)
>
> > *"The only non-BO baseline in the study was SPSA. As can be seen from Figure 3 and Table 2, SPSA performed very poorly in all scenarios and had almost no optimization capability. This seems to lead readers to believe that non-BO methods are not applicable to this task."*
>
> **Response:**
> We appreciate this feedback and would like to clarify the rationale for including SPSA as the non-BO baseline:
>
> - SPSA (Simultaneous Perturbation Stochastic Approximation) is a **widely used method in OD estimation**, both in academia [1–5] and in practice [6–8], including projects funded by USDOT and MnDOT.
> - SPSA is attractive due to its simplicity, low evaluation cost, and compatibility with simulation-based optimization tasks.
>
> However, SPSA’s **limitations in high-dimensional and noisy settings are well known** [9–11], including unstable gradient estimates and lack of structural modeling.
>
> To address this in the final version, we will:
> - Explicitly discuss SPSA’s known weaknesses in high-dimensional problems,
> - Mention common variants such as W–SPSA [9], c–SPSA [10], and PC–SPSA [11], and explain why they were not used (e.g., dependence on historical data, weight matrices, or pre-clustering),
> - Emphasize that BO is not the only tool available, but it is among the **most promising sample-efficient** approaches under compute constraints, a critical need in modern OD estimation tasks.
>
> Our goal is not to dismiss non-BO methods but to **spark new directions** for methods that can match BO’s sample efficiency, scalability, and flexibility in high-cost domains.
>
> ---
>
> ## References
>
> [1] Balakrishna, R., Ben-Akiva, M., & Koutsopoulos, H. N. (2007). Offline calibration of dynamic traffic assignment: simultaneous demand-and-supply estimation. Transportation Research Record, 2003(1), 50-58.
>
> [2] Vaze, V., Antoniou, C., Wen, Y., & Ben-Akiva, M. (2009). Calibration of dynamic traffic assignment models with point-to-point traffic surveillance. Transportation Research Record, 2090(1), 1-9.
>
> [3] Cipriani, E., Florian, M., Mahut, M., & Nigro, M. (2011). A gradient approximation approach for adjusting temporal origin–destination matrices. Transportation Research Part C: Emerging Technologies, 19(2), 270-282.
>
> [4] Ben-Akiva, M. E., Gao, S., Wei, Z., & Wen, Y. (2012). A dynamic traffic assignment model for highly congested urban networks. Transportation research part C: emerging technologies, 24, 62-82.
>
> [5] Ros-Roca, X., Montero, L., Barceló, J., & Nökel, K. (2021, June). Dynamic origin-destination matrix estimation with ICT traffic measurements using SPSA. In 2021 7th International Conference on Models and Technologies for Intelligent Transportation Systems (MT-ITS) (pp. 1-8). IEEE.
>
> [6] Galgano, S., Talas, M., Opie, K., Marsico, M., Weeks, A., Wang, Y., ... & Muthuswamy, S. (2021). Connected vehicle pilot deployment program phase 1: Performance measurement and evaluation support plan: New york city (No. FHWA-JPO-16-302). United States. Department of Transportation. Intelligent Transportation Systems Joint Program Office.
>
> [7] Talas, M., Opie, K., Gao, J., Ozbay, K., Yang, D., Rausch, R., ... & Sim, S. (2021). Connected Vehicle Pilot Deployment Program Phase 3–System Performance Report-New York City (No. FHWA-JPO-18-715). United States. Department of Transportation. Intelligent Transportation Systems Joint Program Office.
>
> [8] Ban, J., Angah, O., Zhang, Y., & Guo, Q. (2022). A multiscale simulation platform for connected and automated transportation systems.
>
> [9] Antoniou, C., Azevedo, C. L., Lu, L., Pereira, F., & Ben-Akiva, M. (2015). W–SPSA in practice: Approximation of weight matrices and calibration of traffic simulation models.
>
> [10] Tympakianaki, A., Koutsopoulos, H. N., & Jenelius, E. (2015). c-SPSA: Cluster-wise simultaneous perturbation stochastic approximation algorithm and its application to dynamic origin–destination matrix estimation. Transportation Research Part C: Emerging Technologies, 55, 231-245.
>
> [11] Qurashi, M., Ma, T., Chaniotakis, E., & Antoniou, C. (2019). PC–SPSA: Employing dimensionality reduction to limit SPSA search noise in DTA model calibration. IEEE Transactions on Intelligent Transportation Systems, 21(4), 1635-1645.

---

> > ### Comment · Reviewer_wSpt · 2025-08-03
> >
> > I  sincerely thank the authors for their detailed response. I have no further questions and will update my score to the positive.

---

> > > ### Author Response · Authors · 2025-08-03
> > >
> > > Thank you for recognizing our clarifications and for your positive feedback on the paper.

---

### Official Review · Reviewer_zaHb · 2025-07-02

**Rating:** 5
**Confidence:** 3

**Summary:**

The paper provides: a Bayesian optimization benchmarks for urban mobility OD problem. The whole framework includes a simulator based on SUMO and several BO models with the corresponding optimization methods.

**Additional Feedback:**

No.

**Dataset Code Accessibility:**

Yes

**Dataset Code Comments:**

The detailed code is provided on Github with a Readme.

**Ethical Comments:**

No.

**Ethical Considerations:**

No, there are no or only very minor ethics concerns

**Final Justification:**

The authors have addressed all my concerns and have also promised to include, in the final version of the paper, an additional example where OD demand is inferred using link-level speed data instead of traffic counts as ground truth. Therefore, I am raising my original score.

**Limitations Weaknesses:**

1. It is unclear what is the main contribution of this framework. If considering the simulation is the main contribution, then this paper is actually based on an existing simulation framework. The network data is also public available and only extracted around a small area. If considering the BO optimization as the main contribution, then is only includes four methods without too many insights about when BO works well, when BO optimization does not work. If considering combining both aspects as the main contribution, then there is lack of discussion about why BO is necessary for OD exploration, are BO methods better than other popular methods for OD estimation?
2. As described in Section 3.3, the dataset comprises both simulation networks and real sensor data. The simulation network is constructed using the established tool SUMO, while the real-world traffic traces are sourced from the PEMS-BAY dataset. These two data types are integrated through a matching process that aligns with standard practices in traffic analysis. Therefore, the contribution of this dataset is relatively limited in terms of novelty.

**Strengths Contributions:**

1. The code is available on the Github.
2. The simulation with the optimization methods is combined together as a whole workflow for further usage.
3. The workflow is evaluated on several different road networks.

---

> ### Author Rebuttal · Authors · 2025-07-30
>
> ---
>
> We thank you for raising important questions about the motivation, novelty, and contributions of our work. Below, we address your concerns point-by-point and offer additional clarification on the design, intent, and significance of our benchmark.
>
> ---
>
> ## Reviewer Comment 1: What is the main contribution of this framework?
>
> > *“It is unclear what is the main contribution of this framework. If considering the simulation is the main contribution, then this paper is actually based on an existing simulation framework. ...If considering the BO optimization as the main contribution, then it only includes four methods without too many insights about when BO works well, when BO optimization does not work. …”*
>
> **Response:**
> As stated in Lines 60–61 of the submitted paper, the primary contribution of our work lies in bridging high-dimensional Bayesian Optimization (BO) and transportation engineering through a **realistic and reproducible benchmark** for simulation-based OD estimation. This benchmark is built around an industry-grade simulation stack, real sensor data, and modular code for evaluating and extending BO algorithms.
>
> This contribution is threefold:
>
> 1. **Benchmarking a high-impact real-world problem for BO research:**
>    OD estimation is a core challenge in transportation systems, yet existing BO benchmarks rarely address such high-dimensional, structured, and noisy domains. Our work introduces this challenge to the BO community in a reproducible and extensible form. As other reviewers including *Reviewer rZqp* and *Reviewer wSpt* acknowledged, as reflected in the comments below, BO4Mob fills a gap by offering the first standardized benchmark linking BO research to real-world mobility problems and encouraging progress in both areas.
> > - *To the best of my knowledge, this is the first benchmark framework for urban mobility problems with a focus on Bayesian optimisation. I believe it has strong potential to foster research on BO methods in this domain. (comment from *Reviewer rZqp*)*
> > - *This paper bridges the gap between advanced Bayesian optimization (BO) algorithm research and real-world traffic engineering applications, addressing the issue of the lack of a standardized testing platform. (comment from *Reviewer wSpt*)*
> > - *The proposed framework provides a challenging “testing ground” for the research community, promoting progress in related fields, like Bayesian optimization and traffic simulation. (comment from *Reviewer wSpt*)*
>
>
>
> 2. **Practical Integration for Multi-Network Benchmarking:**
> Although SUMO and PeMS-BAY are publicly available, the process of building a functioning benchmark is far from plug-and-play. This level of integration is rare even in the transportation domain: most top-tier transportation papers rely on a single case study due to the difficulty of setup. Setting up multiple networks is particularly time-intensive and uncommon in the field. In contrast, we offer five networks of increasing complexity, allowing for scalability evaluation across dimensions up to 10,100. We reconstructed a multi-scale, multi-network simulation environment for the San Francisco Bay Area, including:
>    - OD pair generation (Appendix A),
>    - Sensor-to-link alignment and filtering based on physical principles (Appendix B),
>    - Ensuring simulation stability and match to real-world sensor dynamics (Section 3.3, L183–187),
>    - Adapting the simulation to work with BO frameworks and optimization loops.
>
> 3. **Constructing a Benchmark with Core Technical Challenges**
> As noted in our response to *Reviewer Ez2g*, BO4Mob is a benchmark designed to expose fundamental technical challenges for Bayesian Optimization in transportation problems. It defines a class of optimization tasks involving structurally complex input spaces with domain-specific spatial and topological dependencies, which standard BO methods struggle to model effectively. The problem is inherently high-dimensional, with no latent low-dimensional structure, as shown in New Empirical Result 1. In addition, noisy and incomplete sensor data distorts the objective function, making BO less efficient and more prone to sub-optimal solutions. These aspects make BO4Mob a challenging and realistic testbed for developing and evaluating structure-aware, noise-robust BO methods.
>
>    We will revise the manuscript to better highlight these emerging insights as potential **research directions** for the BO community, including:
>    - The need for adaptive feature selection as shown by filtering unobservable OD pairs (Appendix F),
>    - The challenge of high simulation cost for BO under tight evaluation budgets,
>    - The difficulty of finding effective priors or kernels for structured domains like traffic.
>
>
> ---
>
> ## Reviewer Comment 2: Why is BO necessary or appropriate for OD estimation?
>
> > *“..., there is a lack of discussion about why BO is necessary for OD exploration. Are BO methods better than other popular methods for OD estimation?”*
>
> **Response:**
> This is a central question and we thank you for the opportunity to elaborate.
>
> OD estimation in large-scale transportation systems is:
> - **High-dimensional** (up to thousands of variables),
> - **Non-differentiable**, due to the stochastic nature of agent-based simulators,
> - **Black-box** and **computationally expensive** (e.g., Full Region simulation takes over 11 hours),
> - **Under-determined**, since multiple OD configurations may explain the same sensor data.
>
> Such characteristics are precisely what BO was designed to handle: it builds a sample-efficient surrogate model, chooses candidate solutions via acquisition functions, and handles noisy, constrained, black-box objectives.
>
> In contrast, traditional methods in the transportation literature rely on SPSA and its variants [1–5], which use noisy gradient approximations and fail to exploit problem structure. While advanced forms like W–SPSA [6], c–SPSA [7], and PC–SPSA [8] improve upon this, they often require:
> - Prior tuning or clustering,
> - Access to historical data,
> - Manual weight matrix design,
> - And still struggle at scale.
>
> Our benchmark shows that **BO consistently outperforms SPSA**, and also identifies cases where BO struggles (e.g., Full Region) — offering concrete directions where BO methods can improve. We will expand this discussion in the revision to clarify:
> - The limitations of non-BO baselines,
> - The specific gaps BO fills in this domain,
> - Why OD estimation is a canonical example of a BO-relevant problem.
>
> ---
>
> ## Reviewer Comment 3: The data and simulation tools are public; is the dataset contribution novel?
>
> > *“... ,the contribution of this dataset is relatively limited in terms of novelty.”*
>
> **Response:**
> We appreciate this concern and agree that simply combining public tools is not sufficient for a benchmark contribution. However, as described above and in Section 3.3, our work is more than an aggregation of existing assets. It is an **engineered, validated, and extensible platform** for BO evaluation in a complex real-world task.
>
> This aligns with the NeurIPS Datasets and Benchmarks Track’s emphasis on **integration, reproducibility, and extensibility** as pillars of benchmark contribution NeurIPS 2021 Benchmark Announcement.
>
> In response to suggestions from *Reviewer rZqp*, we’ve prepared our GitHub documentation to include:
> - How to integrate new BO methods,
> - How to customize GP kernels and acquisition functions,
> - How to run the benchmark with custom network configurations.
>
> These improvements support the long-term impact and community adoption of the benchmark.
>
> ---
>
> We thank you again for the thoughtful and constructive feedback. We will revise the manuscript to:
> - Emphasize the integration effort and novelty of the benchmark,
> - Better justify the use of BO for OD estimation,
> - Highlight key empirical insights about BO performance and future research directions.
>
> ---
>
> **References:**
> [1] Balakrishna, R., Ben-Akiva, M., & Koutsopoulos, H. N. (2007). Offline calibration of dynamic traffic assignment: simultaneous demand-and-supply estimation. Transportation Research Record, 2003(1), 50-58.
>
> [2] Vaze, V., Antoniou, C., Wen, Y., & Ben-Akiva, M. (2009). Calibration of dynamic traffic assignment models with point-to-point traffic surveillance. Transportation Research Record, 2090(1), 1-9.
>
> [3] Cipriani, E., Florian, M., Mahut, M., & Nigro, M. (2011). A gradient approximation approach for adjusting temporal origin–destination matrices. Transportation Research Part C: Emerging Technologies, 19(2), 270-282.
>
> [4] Ben-Akiva, M. E., Gao, S., Wei, Z., & Wen, Y. (2012). A dynamic traffic assignment model for highly congested urban networks. Transportation research part C: emerging technologies, 24, 62-82.
>
> [5] Ros-Roca, X., Montero, L., Barceló, J., & Nökel, K. (2021, June). Dynamic origin-destination matrix estimation with ICT traffic measurements using SPSA. In 2021 7th International Conference on Models and Technologies for Intelligent Transportation Systems (MT-ITS) (pp. 1-8). IEEE.
>
> [6] Antoniou, C., Azevedo, C. L., Lu, L., Pereira, F., & Ben-Akiva, M. (2015). W–SPSA in practice: Approximation of weight matrices and calibration of traffic simulation models.
>
> [7] Tympakianaki, A., Koutsopoulos, H. N., & Jenelius, E. (2015). c-SPSA: Cluster-wise simultaneous perturbation stochastic approximation algorithm and its application to dynamic origin–destination matrix estimation. Transportation Research Part C: Emerging Technologies, 55, 231-245.
>
> [8] Qurashi, M., Ma, T., Chaniotakis, E., & Antoniou, C. (2019). PC–SPSA: Employing dimensionality reduction to limit SPSA search noise in DTA model calibration. IEEE Transactions on Intelligent Transportation Systems, 21(4), 1635-1645."

---

> > ### Comment · Reviewer_zaHb · 2025-08-02
> >
> > I appreciate the authors' response.
> >
> > One question that remains is whether OD can be reliably estimated from other features such as flow, speed or GPS trajectories, and how challenging that approximation is in practice. It would be helpful to clarify under what conditions such approximations fail in real-world scenarios.
> >
> > It will also be helpful to further clarify whether there is any essential difference between the proposed dataset and existing datasets available in the literature—for example, those listed in Table IV of the survey paper "Yin, Xueyan, et al. "Deep learning on traffic prediction: Methods, analysis, and future directions." IEEE Transactions on Intelligent Transportation Systems 23.6 (2021): 4927-4943." or Table 1 of "Wang, Sheng, et al. "A survey on trajectory data management, analytics, and learning." ACM Computing Surveys (CSUR) 54.2 (2021): 1-36".
> >
> > Several real-world traffic datasets already exist, and origin-destination (OD) information can be inferred from other features such as traffic flow/trajectories. Given this, what distinguishes the proposed benchmark from applying feature engineering techniques to existing datasets?

---

> > > ### Author Response · Authors · 2025-08-03
> > >
> > > Thank you for your thoughtful questions and valuable feedback.
> > >
> > > First, we clarify that our task is to infer OD demand from traffic flow (count) data—that is, the number of vehicles passing each sensor location over a fixed time horizon. In the manuscript, we used the term traffic count instead of traffic flow to make the concept more intuitive for readers without a transportation background.
> > >
> > > While OD demand can, in principle, be estimated from other features such as speed or GPS trajectories, in practice, such inference is severely limited by data coverage and quality. GPS trajectory penetration rates in typical agency-level deployments are often low, with median values around 2% and many locations exhibiting rates below 1% [1], and even in the most data-rich urban areas, they rarely exceed 10–15%. Although large technology companies such as Google may collect extensive trajectory data, privacy regulations generally prohibit sharing this information with third parties, including local transportation agencies such as state DOTs, cities, or counties. While this remains a promising research direction (see, e.g., [2]), such data constraints make it difficult to recover reliable, population-wide OD matrices without substantial model-based extrapolation.
> > >
> > > The datasets cited in Wang et al. (2021) are illustrative: T-drive and Porto consist of taxi traces, NYC and Chicago datasets capture ride-hailing trip records (aggregated passenger demand), and several others are microscopic driving datasets intended for behavior analysis—none provide comprehensive population-level OD coverage suitable for large-scale demand estimation.
> > >
> > > Similarly, the datasets in Table IV of Yin et al. (2021) are primarily designed for short-term traffic prediction, where the goal is to predict flow or speed for the next time steps given the past time steps. Even the “demand” datasets in that table adopt this temporal prediction paradigm. Similarly, the datasets in Wang et al. (2021) include T-drive and Porto (taxi traces), NYC and Chicago (ride-hailing trip records, aggregated passenger demand), and several microscopic driving datasets intended for behavior analysis—none provide comprehensive, population-level OD coverage for large-scale demand estimation.
> > >
> > > While we agree that several real-world traffic datasets exist, we respectfully note that the presence of such datasets does not imply that OD information can be reliably inferred from them. OD estimation is a distinct and substantially more challenging task, and our paper addresses precisely this gap by proposing a standard benchmark for OD inference—rather than merely presenting another dataset.
> > >
> > > Our benchmark addresses the inverse problem of OD estimation: inferring a high-dimensional static or quasi-static demand vector from sparse, noisy link-level observations. This formulation, data representation, and evaluation protocol are fundamentally different from those of short-term forecasting benchmarks.
> > >
> > > Moreover, our contribution is not limited to the dataset itself; the benchmark encompasses the OD inference task, the dataset, and an accompanying simulation environment. Constructing such an environment is a highly time-intensive effort, and to the best of our knowledge, it is rare for existing benchmarks to provide multiple subnetworks within a single framework. In fact, many of the datasets listed in Table IV of Yin et al. (2021), which include speed, flow, or travel time data, can potentially be integrated into our simulation environment to support more extensive benchmarking.
> > >
> > > To further demonstrate the flexibility of our benchmark, we will include in the final version of the paper an additional example where OD demand is inferred using link-level speed data (rather than traffic counts) as ground truth. This will showcase that our framework supports OD inference not only from count-based data but also from other types of traffic measurements, such as link speeds.
> > >
> > > Thank you again for your insightful feedback.
> > >
> > > ---
> > >
> > > Reference:
> > >
> > > [1] Markovic, N., Miller, S., Laan, Z. V., & Wang, Y. (2020). Visual exploration of Utah trajectory data and their applications in transportation (No. NITC-SS-1264). National Institute for Transportation and Communities (NITC).
> > >
> > > [2] Choi, S., Kim, J., & Yeo, H. (2021). TrajGAIL: Generating urban vehicle trajectories using generative adversarial imitation learning. Transportation Research Part C: Emerging Technologies, 128, 103091.

---

> > > > ### Comment · Reviewer_zaHb · 2025-08-03
> > > >
> > > > Thank you for the clarification. I have no further concerns and will raise my score accordingly.

---

> > > > > ### Author Response · Authors · 2025-08-03
> > > > >
> > > > > Thank you for your positive evaluation of our paper. We also appreciate your acknowledgment of the clarifications we provided.

---

### Official Review · Reviewer_Ez2g · 2025-07-02

**Rating:** 4
**Confidence:** 3

**Summary:**

This paper proposes a Bayesian Optimization (BO) benchmark for the origin-destination (OD) travel estimation problem in urban road networks. Specifically, the task is to optimize the OD estimations between OD pairs, such that the simulated traffic results can maximally match with the observed statistics from real-world sensors. The authors divide the problem into 5 difficulty levels according to the network size, and validate their datasets with common BO baselines such as SAASBO and TuRBO. The results show that the simulation results from BO-optimized OD estimations align better with the ground truth compared to results from the standard SPSA method.

**Additional Feedback:**

1. The task description in the introduction can be further improved. Although the authors listed many examples in the introduction, I am afraid it is a bit hard for the audience without an urban design background (like me) to quickly understand the role of BO in this problem.  That is, what is the objective function and the configurations to optimize? However, it seems that the authors only make this clear until Page 4, i.e., minimizing the difference between ground truth and simulation w.r.t. the OD estimation.

2. I think the experiment part will also benefit from adding a section for GP validation. That is, how well different GPs (e.g., RBF kernel or Mattern kernel) fit the data, as the high-dimensional limitation of BO largely stems from the GP.

3. Is the paper under a noisy observation setting? In particular, how do you obtain the simulation result $y_i^{sim}(x)$ given an OD estimation x? Do you simulate once, or simulate multiple times and then take the average?

4. It would be good to have a short description of BO with mathematical definitions in your methodology (or preliminary).

5. (Minor) Add a random sample baseline in the experiments if possible.

**Dataset Code Accessibility:**

Yes

**Dataset Code Comments:**

The benchmark is hosted on a GitHub repo, in which the authors provided a detailed README.md for running their code.

**Ethical Considerations:**

No, there are no or only very minor ethics concerns

**Final Justification:**

The author's rebuttal addressed my concerns, and I increased my score to borderline accept.

**Limitations Weaknesses:**

Since NeurIPS is a machine learning venue, my main concern of this work is about its benchmarking value for BO literature: except for its practical motivation, how does BO4Mob distinguish itself from the existing benchmarks?

To the best of my knowledge, some common benchmarks, e.g. ROVER (60D) [1], MOPTA08 (124D) [2], Lasso-DNA (180D) [3], SVM (388D) [2], Ant (888D) and Humanoid (6392D) [5], have been used in the high-dimensional BO literature [1, 2, 4, 5, 6]. Does BO4Mob provide new technical challenges that are overlooked by the literature?

For example, as pointed out by the authors in Section 2.1, one recent work [5] found that Lasso-DNA and Mopta08 are not truly high-dimensional by looking at the length scale at different dimensions, in which many input variables seem to have little influence on the objective. How does BO4Mob behave in this case? Is there any empirical evidence that shows the importance of each input feature (i.e., each OD pair) to the objective?

While the authors briefly mentioned some prior works in the related work section, a more detailed discussion will greatly help persuade the audience of its technical contribution.

With that being said, I am still very happy to see a new benchmark for BO driven by a real-world application. Please see my additional comments & questions in the following section, and I am open to further discussion in the rebuttal.


Reference

[1] Scalable Global Optimization via Local Bayesian Optimization. NeurIPS-2019

[2] High-dimensional Bayesian optimization with sparse axis-aligned subspaces. UAI-2021

[3] LassoBench: A High-Dimensional Hyperparameter Optimization Benchmark Suite for Lasso. AutoML-2022

[4] Vanilla Bayesian Optimization Performs Great in High Dimensions. ICML-2024

[5] Understanding High-Dimensional Bayesian Optimization. ICML-2025

[6] Standard Gaussian Process is All You Need for High-Dimensional Bayesian Optimization. ICLR-2025

**Strengths Contributions:**

(1) This paper introduced a new application field for BO, where the task is derived from a real-world optimization problem that is inherently black-box and expensive to evaluate. Additionally, when increasing the number of OD pairs, the benchmark becomes high-dimensional naturally, and is thus suitable for testing high-dimensional BO algorithms.

(2) The paper is well-structured overall, with good demonstrations of the urban simulation problem to help readers quickly understand the underlying problem.

---

> ### Author Rebuttal · Authors · 2025-07-30
>
> ---
>
> We thank you for your insightful and constructive feedback. We agree that a more detailed discussion of related works and new empirical insights will help clarify the technical contributions of our benchmark.
>
> ---
>
> ## New Empirical Result 1:
>
> > *""...,one recent work [5] found that Lasso-DNA and Mopta08 are not truly high-dimensional …, in which many input variables seem to have little influence on the objective. How does BO4Mob behave in this case? ...""*
>
> **Response:**
>
> To examine the influence of each input variable, we conducted an analysis using the method proposed in [5], which identifies dominant and secondary variables based on input sensitivity.
>
> We found that, unlike Lasso-DNA and Mopta08, BO4Mob problems tend to have a high ratio of influential input variables. For example, in the Small Region network with 151 OD variables, over 95% (144.3 on average) were identified as dominant. This supports the view that BO4Mob captures truly high-dimensional and complex optimization settings, where most input features cannot be ignored.
>
> | Network| # dom. | # sec. |
> | - | - | - |
> | Simple Ramp| 3| 0|
> | One-Way Corridor | 19.3| 1.7|
> | Junction| 35.6| 8.4|
> | Small Region| 144.3| 6.7|
>
> ---
>
> ## New Empirical Result 2:
>
> > *""I think the experiment part will also benefit from adding a section for GP validation. ...""*
>
> **Response:**
> We agree that understanding how well different GP kernels fit the objective function is valuable, especially given the known limitations of GPs in high-dimensional settings. To investigate this, we present two new empirical results below. **Additionally, to support ease of diverse experiments for future users, we will provide detailed documentation and example scripts** for customizing Gaussian Process (GP) settings after the final decision, allowing users to investigate how different GP configurations—such as kernel choice and hyperparameter tuning—affect optimization performance.
>
> First, we conducted an experiment where we split the available data (archived from previous BO runs), consisting of OD inputs and corresponding loss values, in half and used the first half to fit a GP model with different kernels. We compared three commonly used kernels: Matern 1.5, Matern 2.5 (which we used in our experiments), and RBF. Then, we evaluated the negative log predictive density (NLPD) for each of the four networks. The table shows the average NLPD values (over two runs) for each network. Bold values indicate the lowest (i.e., best) scores.
>
> | Network | Matern 1.5 | Matern 2.5 | RBF|
> |-|-|-|-|
> | Simple Ramp| -2.477 | **-2.564** | -2.388 |
> | One-Way Corridor |-1.248 | -1.238 | **-1.288**|
> | Junction  | -2.188 | **-2.190** | -2.159|
> | Small Region |-2.866 |**-2.875**|-1.768|
>
> These results suggest that while all three kernels perform reasonably well, kernel choice can meaningfully affect GP fit quality, and Matern kernels generally offer improved performance in our settings.
>
> Following the GP validation, we conducted an additional experiment comparing the optimization performance of each kernel within the BO4Mob framework. Due to computational constraints, the Junction scenario was only run twice; however, the results across networks still provide meaningful insights into kernel-dependent performance. Full experimental details and results will be included in the appendix after the final decision.
>
> | Network | Vanilla BO (Matern1.5) | Vanilla BO (Matern2.5) | Vanilla BO (RBF) | TuRBO (Matern1.5) | TuRBO (Matern2.5) | TuRBO (RBF) |
> |-|-|-|-|-|-|-|
> | Simple Ramp | 0.0011| 0.0033| 0.0074| **0.0002** | 0.0007| 0.0012|
> | One-Way Corridor | 0.0981| 0.1055 | 0.1121| **0.0840** | 0.0898| 0.0934|
> | Junction| **0.2300**| 0.2326 | 0.2303| 0.2380| 0.2337| 0.2323|
>
>
> ---
>
> ## New Empirical Result 3:
>
> > *""Add a random sample baseline in the experiments...""*
>
> **Response:**
> We agree that including a random sampling baseline provides a valuable point of comparison. Accordingly, we have added a random search baseline using Sobol sequences within network-specific OD bounds with a similar sample size to other BO models. Improvement values in the brackets represent the relative improvement of BO Best over random search.
>
>  | Network | Initial Solution (Avg / Min) | Random Search | BO Best |
> | - | - | - | - |
> | Simple Ramp| 0.396 / 0.167| 0.071 (110 runs) | 0.001 [99.02%]|
> | One-Way Corridor | 0.524 / 0.316| 0.145 (320 runs) | 0.070 [51.54%]|
> | Junction| 0.636 / 0.509| 0.436 (830 runs) | 0.233 [83.89%]|
> | Small Region | 0.882 / 0.847 | 0.598 (3050 runs) | 0.258 [56.84%]|
>
> These results confirm that BO significantly outperforms random sampling across all benchmark settings.
>
> ---
>
> ## Reviewer Comment 1:
>
> > *""Since NeurIPS is a machine learning venue, my main concern of this work is about its benchmarking value for BO literature: except for its practical motivation, how does BO4Mob distinguish itself from the existing benchmarks? … Does BO4Mob provide new technical challenges that are overlooked by the literature?""*
>
> **Response:**
> We clarify that **BO4Mob introduces distinct technical challenges** not addressed by existing high-dimensional benchmarks:
>
> 1. **Leveraging Structured, Domain-Specific Priors in BO**
> BO4Mob presents distinct technical challenges beyond standard high-dimensional settings. In OD estimation, the input variables exhibit strong spatial and topological dependencies driven by the transportation network, resulting in complex interactions that standard BO methods with simple kernels fail to capture. These challenges point to the need for physics-informed priors and domain-specific surrogate models that incorporate transportation network structure into GP kernels. Embedding such structure can reduce acquisition flatness and enable more effective sampling. BO4Mob serves as a benchmark to explore and evaluate these structure-aware, physics-informed BO directions, as discussed in L298–307.
>
> 2. **Inherently High-Dimensional Space**
> For large networks, OD estimation is inherently high-dimensional. As demonstrated in New Empirical Result 1, the problem does not lie in a latent lower-dimensional space. Hence, this benchmark will enable the BO community to further investigate classical challenges that arise in high-dimensional BO, such as vanishing gradients of GP likelihood functions or acquisition functions.
>
> 3. **Noisy Sensor Data and its Impact on Bayesian Optimization**
> Due to measurement errors and missing data, the sensor inputs used to evaluate the objective function are noisy. This compromises the accuracy of BO, making it challenging to recover the true OD matrix. This noise distorts the objective function landscape, hindering BO's efficiency and potentially leading to sub-optimal solutions.
>
> Beyond these technical contributions, BO4Mob also responds to the growing demand in the BO community for real-world, commercially grounded black-box systems. This work was motivated by conversations with BO researchers, including feedback received through venues such as the NeurIPS Workshop on Bayesian Decision-Making and Uncertainty (BDU) and the META Adaptive Experimentation Workshop. An earlier version of this work was presented at the NeurIPS BDU Workshop, where several researchers expressed the need for realistic, domain-specific benchmarks of this nature.
>
> ---
>
> ## Reviewer Comment 2:
>
> > *""..., a more detailed discussion will greatly help persuade the audience of its technical contribution.""*
>
> **Response:**
> We agree that further elaboration on related works would help clarify the technical contributions of our benchmark. We will expand the discussion by adding more details on the empirical results and the technical challenges described above.
>
> ---
>
> ### Reviewer Additional Feedback 1:
>
> > *""..., I am afraid it is a bit hard for the audience without an urban design background (like me) to quickly understand the role of BO in this problem... ""*
>
> **Response:**
> We agree that the task description in the introduction should be clearer, especially for readers without a background in urban transportation. We will revise the introduction to explicitly state that each objective function calls refers to estimating the OD matrix by minimizing the difference between simulated and observed traffic counts and the goal is to find the optimal OD demand vector **$\mathbf{x}$ that optimizes this objective.** This clarification will improve the accessibility and motivation of the paper.
>
> ---
>
> ## Reviewer Additional Feedback 2:
>
> > *""Is the paper under a noisy observation setting? ... Do you simulate once, or simulate multiple times and then take the average?""*
>
> **Response:**
> **Yes**, the paper is set in a noisy observation setting. In our experiments, we simulate **once per OD input vector $\mathbf{x}$**, as the stochasticity introduces minimal variance in output.
>
> To quantify this:
>
> **(1) Vehicle routing randomness**
> We ran 100 repeated simulations per network using a fixed OD input specific to that network. The coefficient of variation of the resulting losses was small — even in the worst case (Small Region), it was only 0.00995.
>
> **(2) Internal dynamics randomness**
> We also varied the simulation seed and observed that results remained stable up to the fifth decimal place.
>
> ---
>
> ## Reviewer Additional Feedback 3:
>
> > *""It would be good to have a short description of BO with mathematical definitions in your methodology…""*
>
> **Response:**
> We agree that a brief explanation of BO would improve the clarity and accessibility of the paper. In the revised version, we will include a concise description of BO, including the formal optimization objective, Gaussian Process modeling, and acquisition function design.
>
> ---
>
> We thank you once again for your detailed comments and suggestions. We believe the revisions and additional analyses will improve the clarity, rigor, and overall quality of the paper. We welcome any further discussion during the rebuttal phase.

---

> > ### Comment · Reviewer_Ez2g · 2025-08-01
> >
> > I would like to thank the authors for their detailed rebuttal. I have no further questions and will update my score to the positive.

---

> > > ### Author Response · Authors · 2025-08-01
> > >
> > > We sincerely appreciate your acknowledgment of our clarifications and your positive evaluation of our paper.

---

### Official Review · Reviewer_rZqp · 2025-07-02

**Rating:** 5
**Confidence:** 3

**Summary:**

The paper proposes a new benchmark framework for Bayesian optimisation in urban mobility problems. The framework includes five road network instances of increasing complexity, as well as four baseline methods representing a range of approaches to high-dimensional black-box optimisation, and a built-in function for visualising the results.

**Dataset Code Accessibility:**

No

**Ethical Considerations:**

No, there are no or only very minor ethics concerns

**Limitations Weaknesses:**

Since the aim of the framework is to bridge the fields of black-box optimisation and transportation engineering, I think it would be helpful to provide a clearer explanation of how a new Bayesian optimisation method can be added to the pipeline. This would make it easier to compare new approaches with the provided baselines and better leverage the framework. Currently, this part seems somewhat unclear.

**Strengths Contributions:**

To the best of my knowledge, this is the first benchmark framework for urban mobility problems with a focus on Bayesian optimisation. I believe it has strong potential to foster research on BO methods in this domain. The framework

---

> ### Author Rebuttal · Authors · 2025-07-30
>
> ---
>
> We thank you for the thoughtful feedback and constructive suggestions. We are especially encouraged by your recognition that this is the first benchmark framework for urban mobility problems with a focus on Bayesian Optimization (BO), and that it has strong potential to advance BO research in this domain.
>
> ---
>
> ## Reviewer Comment 1: Pipeline Clarity for Integrating New BO Methods
>
> > *"Since the aim of the framework is to bridge the fields of black-box optimization and transportation engineering, I think it would be helpful to provide a clearer explanation of how a new Bayesian optimisation method can be added to the pipeline. This would make it easier to compare new approaches with the provided baselines and better leverage the framework. Currently, this part seems somewhat unclear."*
>
> **Response:**
> We completely agree that providing clear guidance for integrating new BO methods is essential for usability and broader adoption.
>
> - While we have not yet updated the code or documentation due to the NeurIPS policy during the review phase, we have prepared a set of detailed instructions that will be released after the review period via our GitHub repository.
> - In brief, new BO methods can be integrated by creating a new class that inherits from the `BaseStrategy` class defined in `./src/optimizers/base_strategy.py`. All optimization strategies used in our paper (e.g., TuRBO, SAASBO) follow this structure, which promotes consistency and modularity.
> - Additionally, in response to *Reviewer Ez2g*'s related comment regarding Gaussian Process customization, we will also add clear inline documentation on how to modify the kernel and other GP-related settings.
> - Additionally, we will include example templates and annotated scripts to demonstrate how new methods can be plugged into the pipeline while reusing the benchmark’s simulation engine, evaluation metrics, and logging infrastructure.
>
> We believe these enhancements will make BO4Mob a user-friendly, extensible testbed for the BO community.
>
> ---
>
> ## Reviewer Comment 2: Response Regarding Dataset Code Accessibility
>
> We would like to kindly clarify that we have indeed released our dataset and code. If there is anything unclear or if further details would be helpful, we are happy to provide them.
>
> ---
>
> We thank you again for recognizing both the technical novelty and practical relevance of our benchmark. We believe BO4Mob fills a critical gap in the high-dimensional BO literature by offering a realistic, scalable, and open-source testbed grounded in operational transportation systems. We welcome further discussion during the rebuttal phase.

---

### Note · Authors · 2025-08-15

We are grateful for the constructive feedback from the reviewers, which helped strengthen and clarify our work. BO4Mob is the first benchmark framework for Bayesian optimization (BO) addressing origin-destination (OD) estimation challenges in urban mobility, offering realistic, high-dimensional, and noisy settings grounded in operational transportation systems. It integrates SUMO simulations with real PeMS sensor data, spans five networks of increasing complexity (up to 10,100 variables), and is fully open-source. After rebuttal and discussion, all reviewers expressed positive views, several scores increased, and no substantive concerns remain.

During the review, we addressed pipeline usability, confirmed the benchmark’s distinctiveness and suitability for BO, and clarified dataset and simulation contributions. We also discussed scenario coverage, including guidelines for adding new networks and the role of the Full Region case as an open challenge for scalable BO research.

With reviewers aligned on BO4Mob’s soundness, novelty, and potential impact, we believe it fills a critical gap between BO research and transportation engineering, providing a challenging, extensible, and realistic testbed for developing advanced optimization methods in complex real-world systems.

---

### Decision · Program_Chairs · 2025-09-18

**Decision:**

Accept (poster)

**Comment:**

The paper introduced BO4Mob, a new high-dimensional Bayesian Optimization (BO) benchmark framework for origin-destination (OD) travel demand estimation as a high-dimensional urban mobility problem.

Reviewer rZqp asked about the pipeline between BO and transportation engineering.
Reviewer Ez2g considers this paper to be well-written. The reviewer's main point is the value of this dataset in the BO lit and elaborated the discussion by comparing it with existing benchmarks. The reviewer is generally satisfied with the author's response.
Reviewer zaHb has a question on the link between OD and BO. The authors answer their fitness and potential new challenges. The reviewer is satisfied with the answers and the promised addition of link-level speed data.
Reviewer WSPT is satisfied by the value of the paper. The questions are mainly on the limitations of the dataset, and the authors' response addresses the concerns.

Since all reviewers are satisfied with the response and this dataset aligns with the value of the D&B track, I recommend acceptance of the paper.

===== FINAL UPDATE FROM DB Track PCs ====

The final decision for this paper has been taken by the program chairs after consultation with the SACs. All Senior Area Chairs have ranked papers according to the feedback from the AC during the review process. We decided to leave the original meta-review to reflect the opinion of the AC in light of the initial discussions with reviewers and SAC.